# Metapopulation ecology links antibiotic resistance, consumption, and patient transfers in a network of hospital wards

Julie Teresa Shapiro[1], Gilles Leboucher[2], Anne-Florence Myard-Dury[3], Pascale Girardo[4], Anatole Luzzati[4], Mélissa Mary[4], Jean-François Sauzon[4], Bénédicte Lafay[5], Olivier Dauwalder[4], Frédéric Laurent[1,4], Gerard Lina[1,4], Christian Chidiac[6], Sandrine Couray-Targe[3], François Vandenesch[1,4], Jean-Pierre Flandrois[5], Jean-Philippe Rasigade[1,4]*

[1]CIRI, Centre International de Recherche en Infectiologie, Université de Lyon, Inserm U1111, Ecole Normale Supérieure de Lyon, Université Lyon 1, CNRS, UMR5308, Lyon, France; [2]Département de Pharmacie, Hospices Civils de Lyon, Lyon, France; [3]Pôle de Santé Publique, Département d'Information Médicale, Hospices Civils de Lyon, Lyon, France; [4]Institut des Agents Infectieux, Hospices Civils de Lyon, Lyon, France; [5]Laboratoire de Biométrie et Biologie Evolutive, UMR CNRS 5558, University of Lyon, Lyon, France; [6]Service des Maladies Infectieuses et Tropicales, Hospices Civils de Lyon, Lyon, France

**Abstract** Antimicrobial resistance (AMR) is a global threat. A better understanding of how antibiotic use and between-ward patient transfers (or connectivity) impact population-level AMR in hospital networks can help optimize antibiotic stewardship and infection control strategies. Here, we used a metapopulation framework to explain variations in the incidence of infections caused by seven major bacterial species and their drug-resistant variants in a network of 357 hospital wards. We found that ward-level antibiotic consumption volume had a stronger influence on the incidence of the more resistant pathogens, while connectivity had the most influence on hospital-endemic species and carbapenem-resistant pathogens. Piperacillin-tazobactam consumption was the strongest predictor of the cumulative incidence of infections resistant to empirical sepsis therapy. Our data provide evidence that both antibiotic use and connectivity measurably influence hospital AMR. Finally, we provide a ranking of key antibiotics by their estimated population-level impact on AMR that might help inform antimicrobial stewardship strategies.

*For correspondence:
jean-philippe.rasigade@univ-lyon1.fr

Competing interests: The authors declare that no competing interests exist.

## Introduction

Antimicrobial resistance (AMR) of pathogenic bacteria progresses worldwide, imposing a considerable burden of morbidity, mortality and healthcare costs (*Laxminarayan et al., 2013*; *Cassini et al., 2019*). AMR is increasingly recognized to emerge in various settings including agriculture (*Johnson et al., 2016*) or polluted environments (*Lübbert et al., 2017*; *Venter et al., 2017*). However, hospitals continue to be important hotspots for AMR in clinically-relevant pathogens (*Chatterjee et al., 2018*; *David et al., 2019*) due to the confluence of strong antibiotic selection pressure, fragile patients, and highly resistant pathogens that can disseminate between wards and facilities through patient transfers (*Safdar and Maki, 2002*; *Snitkin et al., 2012*). These conditions result in hospitals becoming reservoirs of resistant bacteria (*Clarivet et al., 2016*; *Cohen, 1992*; *Pogue et al., 2013*), which can later enter the community (*Huttner et al., 2013*).

The primary hospital-based strategies against AMR are antimicrobial stewardship, which aims to lower the antibiotic pressure, and infection control whose goal is to reduce the transmission of

pathogens (*Manning et al., 2018*). The need for such strategies is widely accepted but because they are often implemented together, the relative importance of each is unclear (*Chatterjee et al., 2018*; *Goff et al., 2017*). The details of how to best implement these strategies are hotly debated (*Lemmen and Lewalter, 2018*), especially regarding which antibiotics should be restricted first as a part of antibiotic stewardship strategies (*Chastain et al., 2018*) or the risk-benefit balance of screening-based patient isolation procedures to reduce transmission (*Kardaś-Słoma et al., 2017*; *Robotham et al., 2016*). Designing effective antibiotic stewardship strategies has been hindered by the paucity of evidence concerning which antibiotics exert the strongest selection pressure. Current rankings of antibiotics for de-escalation and sparing strategies rely on expert consensus with partial agreement (*Weiss et al., 2015*), themselves based on conflicting evidence (*Acar, 1997*; *Huttner et al., 2016*). This is further complicated by the fact that the association between antibiotic use and resistance is not uniform across pathogen species (*Bell et al., 2014*) or classes of antibiotics (*Niehus et al., 2020*). Moreover, antibiotic consumption can also have long-term effects on the carriage of resistant bacteria within patients (*Niehus et al., 2020*).

Understanding the drivers of AMR at the hospital level requires consideration not only of the selection pressure from antibiotics on individual patients, but also of the transmission and dissemination of drug-resistant pathogens in the patient population (*Lipsitch, 2001*; *Lipsitch and Samore, 2002*). However, we lack a quantitative understanding of the respective impacts of selection and transmission on the incidence of AMR infections in hospitals. Observational studies of AMR usually report on the proportion of resistant variants in a limited set of species, which conceals the overall AMR burden and can make interpretation difficult when resistant variants apparently increase in proportion while decreasing in incidence (*Burton, 2009*). Moreover, linking antibiotic use and AMR prevalence at the population level is difficult due to the confounding effects of bacterial transmission and the complexity of the ecological processes underlying AMR (review in *Schechner et al., 2013*). Thus, studies of AMR in hospitals could benefit from ecological frameworks able to simultaneously model the impact of antibiotic use and patient transfers on the incidence of infections with the most relevant pathogens. Metapopulation ecology is such a framework. It was introduced by *Levins, 1969* to explain the persistence of agricultural pests across a set of habitat patches, such as crop fields, and refined by Hanski to account for the characteristics of patches, such as their size, and the connectivity between them (*Hanski, 1998*; *Hanski, 1994*). The metapopulation concept, in which populations of organisms are spread across inter-connected patches with varying characteristics, is useful to describe bacterial pathogens in a hospital network containing wards with different sizes and levels of antibiotic pressure, connected by the transfers of infected patients. Models using the metapopulation framework, beyond their frequent use in wildlife and conservation biology (*Dolrenry et al., 2014*; *Heard et al., 2015*; *MacPherson and Bright, 2011*), have recently provided theoretical grounds for pathogen persistence in the healthcare setting (*Spagnolo et al., 2018*). So far, however, metapopulation models of hospital AMR have been applied to simulated rather than empirical data (*Spagnolo et al., 2018*; *Vilches et al., 2019*).

Here, we used a metapopulation framework to model and isolate the population-level effects of antibiotic use and inter-ward connectivity on the incidence of infections with major pathogen species and their drug-resistant variants within a 357-ward hospital network, using detailed data collected over the course of one year. Our objectives were: (1), to determine the respective impacts of antibiotic use and connectivity on the incidence of infections with resistant pathogens at the population level; and (2), to compare the impacts of the use of specific antibiotics on the ward-level incidence of AMR infections, after controlling for the effect of connectivity. Based on the association patterns between the incidence of 17 pathogen variants and the use of 11 antibiotic classes, our findings highlight both common patterns and species-specific behaviors of pathogens and provide a ranking of key antibiotics by their estimated population-level impact on AMR.

## Results

### Distribution of bacterial pathogens and antibiotic use in a hospital network

We analyzed pathogen isolation incidence in clinical samples, antibiotic use, and patient transfers in 357 hospital wards from the region of Lyon, France. Data for all three measures were collected

during the same period from October 2016 to September 2017. The hospital network contained a total of 4,685 beds. The median ward size was 12 beds (interquartile range, 5 to 20).

Ward-level data were aggregated from 13,915 infection episodes, defined as ward admissions with ≥1 clinical sample positive for *E. coli* or one of the so-called ESKAPE pathogens (*Enterococcus faecium*, *Staphylococcus aureus*, *Klebsiella pneumoniae*, *Acinetobacter baumannii*, *Pseudomonas aeruginosa* and *Enterobacter cloacae* complex), collectively termed ESKAPE$_2$ (*Table 1*). Pathogens were grouped into species-resistance pattern combinations, namely 3$^{rd}$-generation cephalosporin (3GC)-resistant *E. coli*, *E. cloacae* complex and *K. pneumoniae*, carbapenem-resistant *E. coli*, *E. cloacae* complex, *K. pneumoniae*, *P. aeruginosa* and *A. baumannii*, vancomycin-resistant *E. faecium* and methicillin-resistant *S. aureus* (MRSA). Pathogen variants not falling into these resistance groups were collectively referred to as the less-resistant variants (*Table 1*). The median yearly incidence of infection episodes per ward was 24 (interquartile range, 7 to 55).

Infection episodes most frequently involved the less-resistant variants, especially *E. coli*, which were also found in the largest number of wards (*Table 1* and *Figure 1—figure supplement 1*). Resistant variants were consistently less frequent than their less-resistant counterparts in all species (*Figure 1—figure supplement 1*). In enterobacteria (*E. coli*, *K. pneumoniae* and *E. cloacae*), carbapenem-resistant variants were consistently less frequent than 3GC-resistant variants. Infections with vancomycin-resistant *E. faecium* and carbapenem-resistant *A. baumannii* were exceptional, with seven and twelve episodes respectively (*Table 1*).

To estimate the degree of concentration of each variant in the network, we calculated concentration indices defined as the probability that two random occurrences of the same variant originated from the same ward, analogous to the asymptotic Simpson index (see Materials and methods). The concentration index varies from 0% for a uniformly random distribution (each occurrence is in a different ward) to 100% for a maximally concentrated distribution (all occurrences are in the same ward). The concentration of infection episodes was weak (<5%) for all variants, indicating a global lack of clustering (*Table 1*). Concentration increased with resistance (~2-fold increase from the least

**Table 1.** Distribution of ESKAPE$_2$ pathogen infection episodes in 357 hospital wards.

| Species | Resistance profile | Acronym | No. of episodes (%), n = 13,915 | No. of wards (%), n = 357 | Concentration index[a] (%) (95% CI) |
|---|---|---|---|---|---|
| *E. coli* | Susceptible to 3GC and carbapenems | EC | 6,303 (45.3) | 328 (91.9) | 0.6 (0.6, 0.7) |
| | 3GC-resistant | 3GCREC | 737 (5.3) | 207 (58.0) | 0.7 (0.6, 0.8) |
| | Carbapenem-resistant | CREC | 24 (0.2) | 24 (5.6) | 1.4 (0.0, 3.9) |
| *K. pneumoniae* | Susceptible to 3GC and carbapenems | KP | 1,133 (8.1) | 249 (69.7) | 0.7 (0.6, 0.8) |
| | 3GC-resistant | 3GCRKP | 530 (3.8) | 175 (49.0) | 0.9 (0.7, 1.0) |
| | Carbapenem-resistant | CRKP | 43 (0.3) | 32 (9.0) | 1.7 (0.0, 3.5) |
| *E. cloacae* complex | Susceptible to 3GC and carbapenems | EB | 277 (2.0) | 140 (39.2) | 1.0 (0.7, 1.3) |
| | 3GC-resistant | 3GCREB | 212 (1.5) | 116 (32.5) | 0.8 (0.5, 1.0) |
| | Carbapenem-resistant | CREB | 102 (0.7) | 74 (20.7) | 0.7 (0.3, 1.1) |
| *P. aeruginosa* | Carbapenem-susceptible | PA | 1,076 (7.7) | 231 (64.7) | 0.8 (0.7, 0.9) |
| | Carbapenem-resistant | CRPA | 444 (3.2) | 148 (41.5) | 1.5 (1.2, 1.7) |
| *A. baumannii* | Carbapenem-susceptible | AB | 96 (0.7) | 61 (17.1) | 1.3 (0.5, 2.1) |
| | Carbapenem-resistant | CRAB | 12 (0.1) | 10 (2.8) | 3.0 (0.0, 9.8) |
| *E. faecium* | Vancomycin-susceptible | EF | 503 (3.6) | 133 (27.3) | 1.4 (1.2, 1.6) |
| | Vancomycin-resistant | VREF | 7 (<0.1) | 7 (2.0) | 0.0 (0.0, 9.6) |
| *S. aureus* | Methicillin-susceptible | SA | 2,113 (15.2) | 273 (76.5) | 1.1 (1.0, 1.1) |
| | Methicillin-resistant | MRSA | 303 (2.2) | 151 (42.3) | 0.7 (0.5, 0.9) |

NOTE. [a]The concentration index estimates the probability that two episodes taken at random occurred in the same ward. Here we report the concentration index as a percent (0–100%). 3GC, 3$^{rd}$-generation cephalosporins.

to the most resistant variant) in *E. coli*, *K. pneumoniae*, *P. aeruginosa* and *A. baumannii*, suggesting an adaptation of resistant variants to more specific wards within the hospital network compared to their less-resistant, less-concentrated counterparts. This pattern was not found in *E. cloacae* complex, *E. faecium* or *S. aureus*, in which the concentration index remained comparable across resistance categories.

Over the same period (October 2016 to September 2017), antibiotics were prescribed in 86.3% of wards (*Table 2*), with a total consumption of 125.7 defined daily doses per year per bed (ddd/y/b). Antibiotics usually suspected to select for AMR in the selected variants were grouped into 11 classes (*Table 2*). Antibiotics with comparatively rare use (e.g., rifampicin) were excluded. The distribution of antibiotic use in the network was analyzed using the concentration index described above, here representing the probability that two random drug doses were delivered in the same ward. Antibiotic use was diffuse, with concentration indices < 4%, ranging from 0.8% for cefotaxime-ceftriaxone and fluoroquinolones to 3.6% for oxacillin.

## Antibiotic use and connectivity predict the incidence of drug-resistant infections

We used multivariable generalized linear models (GLMs) within the metapopulation framework to disentangle the influences of antibiotic pressure, connectivity, ward size, and ward type on the incidence of infections with the selected pathogens and their resistant variants.

Connectivity quantifies the incoming flux of each pathogen variant in a downstream ward via the transfer of infected patients from upstream wards. Practically, we estimated connectivity for each variant and downstream ward as the sum of the direct transfers from each upstream ward multiplied by the variant's prevalence in that ward (see Materials and methods). Mean connectivity ranged from 168.2 estimated introductions per year for less-resistant *E. coli* to 0.09 for vancomycin-resistant *E. faecium*. Connectivity was always higher for the less-resistant variants compared to the resistant variants, consistent with the higher prevalence of the former (*Supplementary file 1* Table 1a).

Wards were characterized by their size (no. of beds) and type, representing patient fragility. Ward type was coded as one of the following categorical variables: intensive care and blood cancer units, progressive care units, and other wards. 'Other wards' was considered the reference category

**Table 2.** Distribution of the use of 11 antibiotics in 357 hospital wards.

| Antibiotics | Acronym | Prescription volume in ddd/y (%) | No. of wards (%), n = 357 | Concentration index[a] (%), (95% CI) |
|---|---|---|---|---|
| Amoxicillin | AMX | 141,293 (24.0) | 252 (70.6) | 1.8 (1.7, 1.8) |
| Coamoxiclav | AMC | 78,072 (13.3) | 247 (69.3) | 1.0 (0.9, 1.0) |
| First- and second- generation cephalosporins | 1GC/2GC | 12,915 (2.2) | 191 (53.5) | 1.2 (1.1, 1.3) |
| Non-antipseudomonal 3GCs, cefotaxime and ceftriaxone | CTX/CRO | 53,406 (9.1) | 259 (72.5) | 0.8 (0.8, 0.9) |
| Antipseudomonal 3GCs, ceftazidime and cefepime | CTZ/FEP | 29,204 (5.0) | 184 (51.5) | 1.9 (1.8, 1.9) |
| Piperacillin-tazobactam | TZP | 27,593 (4.7) | 198 (55.5) | 1.9 (1.7, 1.9) |
| Carbapenems | IPM/MEM | 25,093 (4.3) | 204 (57.1) | 1.5 (1.4, 1.6) |
| Oxacillin | OXA | 12,374 (2.1) | 143 (40.1) | 3.6 (3.1, 3.7) |
| Vancomycin and teicoplanin | VAN/TEC | 25,376 (4.3) | 206 (57.7) | 1.5 (1.4, 1.5) |
| Fluoroquinolones | FQ | 52,549 (8.9) | 249 (69.7) | 0.8 (0.8, 0.8) |
| Aminoglycosides | AMIN | 12,745 (2.2) | 207 (58.0) | 1.9 (1.5, 1.9) |
| All antibiotics[b] | - | 589,014 (100) | 308 (86.3) | 0.8 (0.8, 0.8) |

NOTE. [a]The concentration index estimates the probability that two antibiotic ddds taken at random were prescribed in the same ward. Here we report the concentration index as a percent (0–100%). [b]Total consumption of systemic-use antibiotics (ATC class J01) including those not considered in the 11 specific drug groups. 3GC, 3rd-generation cephalosporin; ddd, defined daily dose.

in all models. In several pathogen variants, namely carbapenem-resistant *E. coli*, carbapenem-resistant *A. baumannii*, and vancomycin-resistant *E. faecium*, the small sample size in one or several ward categories prevented the inclusion of ward type as a model covariate.

We also considered that the distribution of infections across wards was a source of bias that required a specific adjustment procedure (see Materials and methods). The local prevalence of specific infections (e.g., respiratory tract infections) in a ward influences both the antibiotic use and the observed incidence of infections with a given pathogen, which might confound the relationship between antibiotic use and incidence (*Figure 1—figure supplement 2*). However, the distribution of infections would be difficult to represent as an adjustment covariate with a sufficiently small number of categories. We used a proxy method to circumvent this issue. We assumed that the distribution of infections directly influences the frequency and specimen types (e.g., respiratory vs. urinary tract specimens) of microbiological samples in each ward. Under this assumption, we replaced the unrepresentable distribution of infections with a proxy variable summarizing the distribution of microbiological samples. This proxy variable, which we refer to as the incidence control, was defined as the ward-level incidence of a pathogen variant predicted by patterns of microbiological sampling alone. As expected, the incidence control correlated with both antibiotic use and the incidence of infections in all prevalent variants (*Figure 1—figure supplements 3* and *4*). Of note, the incidence of each variant also correlated with both antibiotic use (*Figure 1—figure supplement 5*) and connectivity (*Figure 1—figure supplement 6*) in bivariate analyses, except for the very rare *A. baumannii* and *E. faecium* resistant variants. Hereafter, all models included the incidence control covariate to adjust for the confounding effect of the distribution of infections.

The incidence of each pathogen variant was modeled in a separate multivariable quasi-Poisson GLM (*Figure 1*). In these GLMs, global antibiotic use was associated with infection incidence in seven pathogen variants independent of connectivity, ward size, and ward type (*Figure 1*), including five resistant variants (3GC-resistant *E. coli*, 3GC-resistant *K. pneumoniae*, carbapenem-resistant *K. pneumoniae*, 3GC-resistant *E. cloacae*, and carbapenem-resistant *E. cloacae*) and two less-resistant variants (*P. aeruginosa* and *E. faecium*). The largest effect size was found in carbapenem-resistant *K. pneumoniae*, in which every doubling of antibiotic use predicted a 47% increase in incidence (95% confidence interval, 19% to 90%).

Connectivity predicted a higher incidence in all variants of *P. aeruginosa* and *E. faecium*, with a stronger effect size in the resistant variants (11.5 and 43.7%, respectively) compared to their less-resistant counterparts (6.1 and 15.3%). A significant association with connectivity was also found for the less-resistant *E. coli*, although with a much smaller effect size (2.6%).

Ward characteristics only weakly predicted infection incidence compared to antibiotic use and connectivity. Ward type, or patient fragility, predicted incidence in several variants, although with large uncertainty margins. Interestingly, associations of incidence with intensive care and blood cancer units were negative (in *E. coli*, 3GC-resistant *E. coli*, 3GC-resistant *E. cloacae*, and MRSA) while associations with progressive care units were positive (in 3GC-resistant *K. pneumoniae* and *P. aeruginosa*). Ward size did not predict incidence in any variant.

## Do associations between antibiotic use and resistance represent AMR selection?

The metapopulation models illustrated in *Figure 1* identified positive associations between total antibiotic use in hospital wards and increased incidences of infections with the more resistant variants of several species. Yet, a correlation with AMR does not necessarily establish a selective role of antibiotics. For instance, a high incidence of resistant infections in a ward can increase antibiotic use through prolonged or combined therapies (*Schechner et al., 2013*). Conversely, the prescription of antibiotics always inactive against a variant is unlikely to be motivated by this variant's incidence and such antibiotics are more likely to provide a direct benefit to the resistant variant. Based on this rationale, we propose stringent criteria to identify whether an association between the use of an antibiotic and the incidence of a variant possibly reflects AMR selection (hereafter, possibly selective associations; see Materials and methods). Under the hypothesis that antibiotic use is either a consequence of AMR or spuriously correlated with AMR, the strength of an association between the use of an antibiotic and the incidence of a variant should not depend on whether the association fulfills the criteria for possible selection.

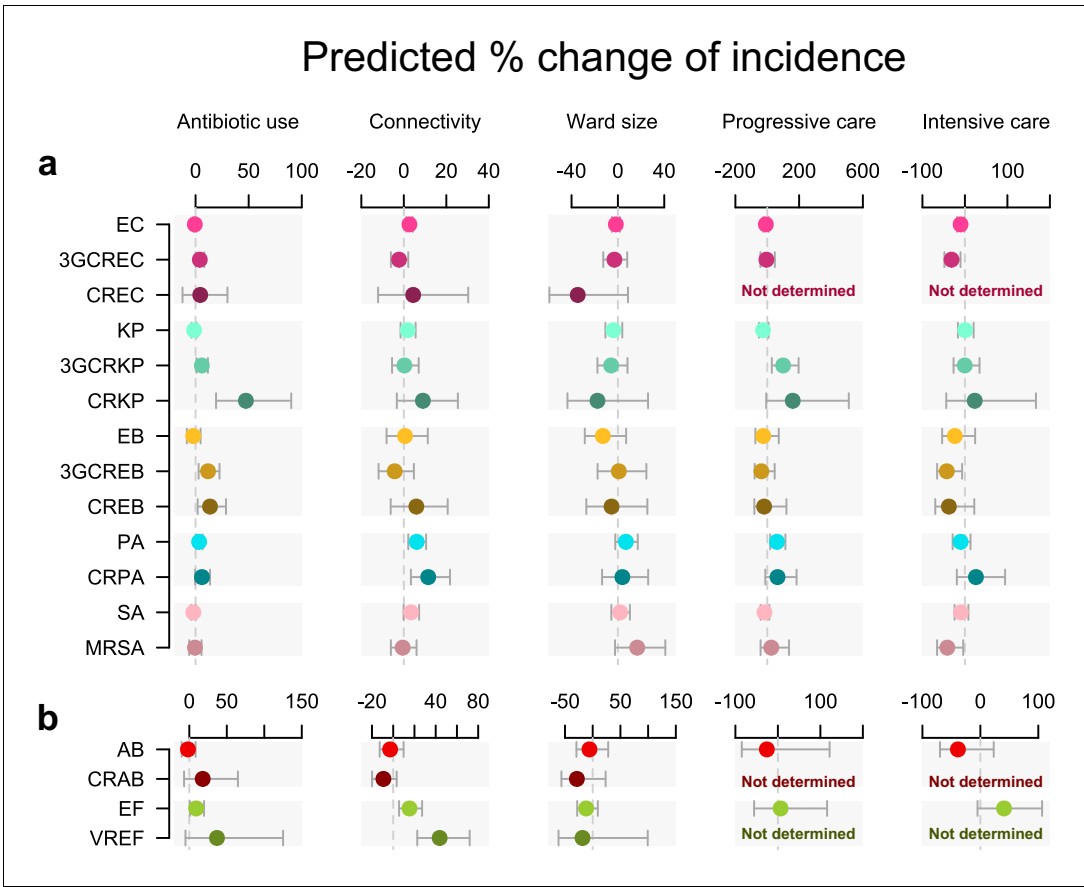

**Figure 1.** Antibiotic use and connectivity predict the incidence of infection with ESKAPE$_2$ pathogen variants. Shown are the predicted percent changes in incidence (points) with 95% confidence interval (bars) for each variant in each ward (n = 357) for every doubling of antibiotic use, connectivity (estimated no. of patients infected with the same variant entering the ward), ward size (no. of beds), and ward type. All models were multivariable quasi-Poisson regressions that included the incidence control covariate (see Materials and methods). Models involving *A. baumannii* and *E. faecium*, which exhibited larger 95% confidence intervals due to smaller incidence of the resistant variants, are shown with separate scales (panel **b**) for readability. In models of CREC, CRAB, and VREF incidence, small sample size in at least one ward category prevented the inclusion of ward type as a covariate and the estimation of the coefficient, marked as 'not determined'. Variant acronyms are listed in *Table 1*.
The online version of this article includes the following figure supplement(s) for figure 1:

**Figure supplement 1.** Incidence of each variant during the study period.
**Figure supplement 2.** Directed acyclic graph representation of causal assumptions.
**Figure supplement 3.** Correlation of ward-level incidence control values with infection incidence in ESKAPE$_2$ pathogen variants.
**Figure supplement 4.** Correlation of incidence control values with observed ward-level antibiotic consumption.
**Figure supplement 5.** Correlation of ward-level antibiotic consumption and infection incidence in ESKAPE$_2$ variants.
**Figure supplement 6.** Correlation of ward-level connectivity and infection incidence in ESKAPE$_2$ variants.

To test this hypothesis, we identified possibly selective associations in our data and we examined whether they were equally likely to be positive compared to other associations. We constructed quasi-Poisson multivariable regression models where the total antibiotic use was replaced with the use of specific antibiotics, along with the incidence control and connectivity covariates (*Figure 2a and b*). Ward type and size, which were weak predictors of incidence in *Figure 1* models, were excluded to avoid introducing additional noise. The 17 variant-specific models each included all antibiotic groups as predictors, including antibiotic classes not expected to exert any direct selection pressure on the variant, such as aminoglycosides on *E. coli*. The models yielded 187 coefficients of

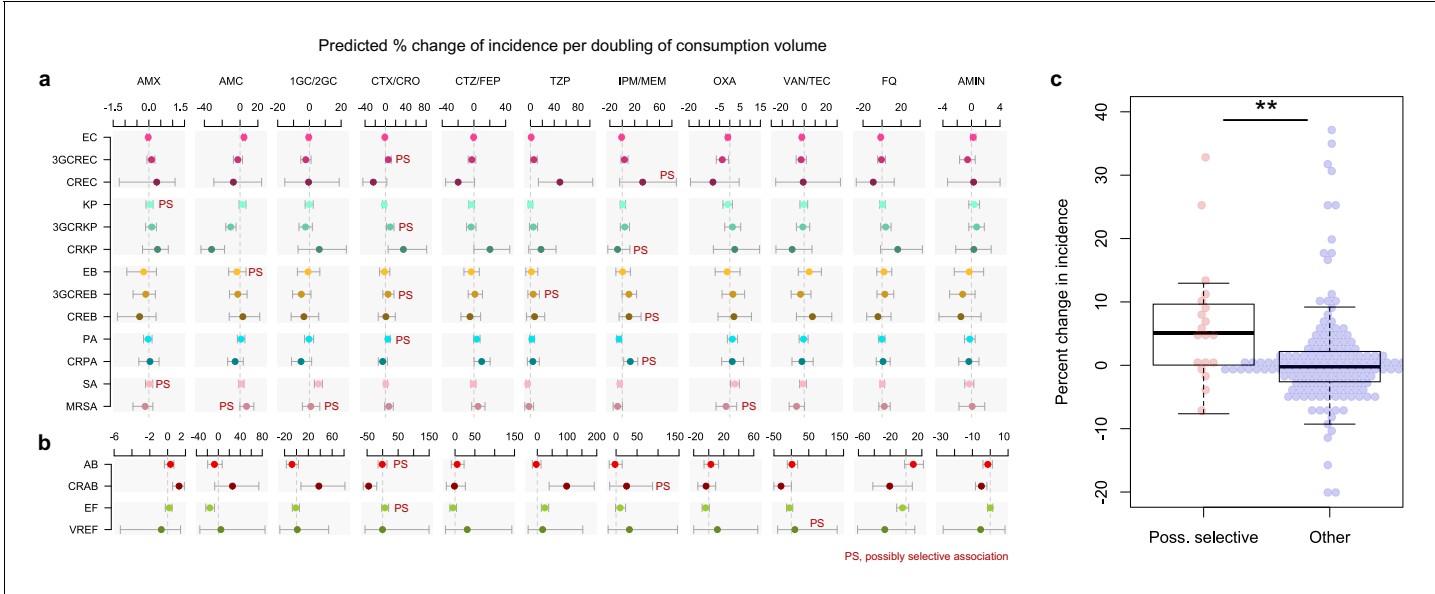

**Figure 2.** Possibly selective associations between the use of specific antibiotics and the incidence of infection with ESKAPE$_2$ pathogen variants. Shown are the predicted percent changes in incidence (points) with 95% confidence interval (bars) for each variant in each ward (n = 357) for every doubling in the consumption volume of 11 antibiotic groups, based on multivariable quasi-Poisson regression models of the incidence of each variant in each ward (n = 357) that included the connectivity and incidence control covariates (see Materials and methods). Associations classified as possibly selective (n = 19) are indicated by a 'PS' mark. Models involving *A. baumannii* and *E. faecium*, which exhibited larger 95% confidence intervals due to smaller incidence of the resistant variants, are shown with separate scales (panel **b**) for readability. (**c**), possibly selective associations had higher coefficients compared to other associations. The center line indicates the median; box limits indicate the upper and lower quartiles; whiskers indicate the 1.5x interquartile range; points indicate the individual coefficients. \*\*p<0.01, two-sided Mann–Whitney *U*-test. Acronyms of pathogen variants and antibiotics are listed in *Tables 1* and *2*, respectively.

which 19 (10.2%) represented possibly selective associations (see Materials and methods). The mean percent change in incidence for every doubling of consumption volume in possibly selective associations was 6.5%, compared to 0.9% in other associations (95% CI of the difference of the means, 0.6 to 10.6%). Three of the four strongest possibly selective associations involved cefotaxime-ceftriaxone, which selected for 3GC-resistant *E. coli*, *K. pneumoniae* and *P. aeruginosa*; the other involved carbapenems selecting for carbapenem-resistant *P. aeruginosa*. Overall, the larger magnitude of the coefficients of possibly selective associations suggests that the local, ward-level selection of drug-resistant variants by antibiotics is measurably pervasive throughout our hospital network.

## Quantifying the drivers of resistance to first-line sepsis therapy

From a clinical standpoint, the most immediate consequence of AMR is the failure to control sepsis with empirical antibiotics, mainly carbapenems and the non-antipseudomonal 3GCs cefotaxime and ceftriaxone. Because such failure can equally result from acquired or intrinsic resistance, the incidence of intrinsically resistant pathogens such as *E. faecium* is of equal clinical importance as that of pathogen variants with acquired resistance mechanisms. To examine the impact of antibiotics on both intrinsic and acquired resistance, we modeled the cumulative incidence of infections with 3GC- and/or carbapenem-resistant variants of the selected pathogens (see Materials and methods).

In these models, antibiotic use was the strongest predictor of the incidence of both carbapenem-resistant (6.5% increase in incidence for every doubling of consumption volume, 95% CI, 2.5 to 11.0%) and 3GC-resistant infections (5.1% increase, 95% CI, 2.7 to 7.5%; *Figure 3a*). Connectivity better predicted the incidence of carbapenem-resistant infections (4.8%, 95% CI, −0.4 to 10.7%) compared with 3GC-resistant infections (1.0%, 95% CI, −2.3 to 4.5%), in line with the comparatively stronger association of connectivity with the incidence of individual carbapenem-resistant variants (*Figure 1*). Ward size had no measurable effect in either model. Carbapenem-resistant infections were not associated with ward type, while 3GC-resistant infections were positively associated with

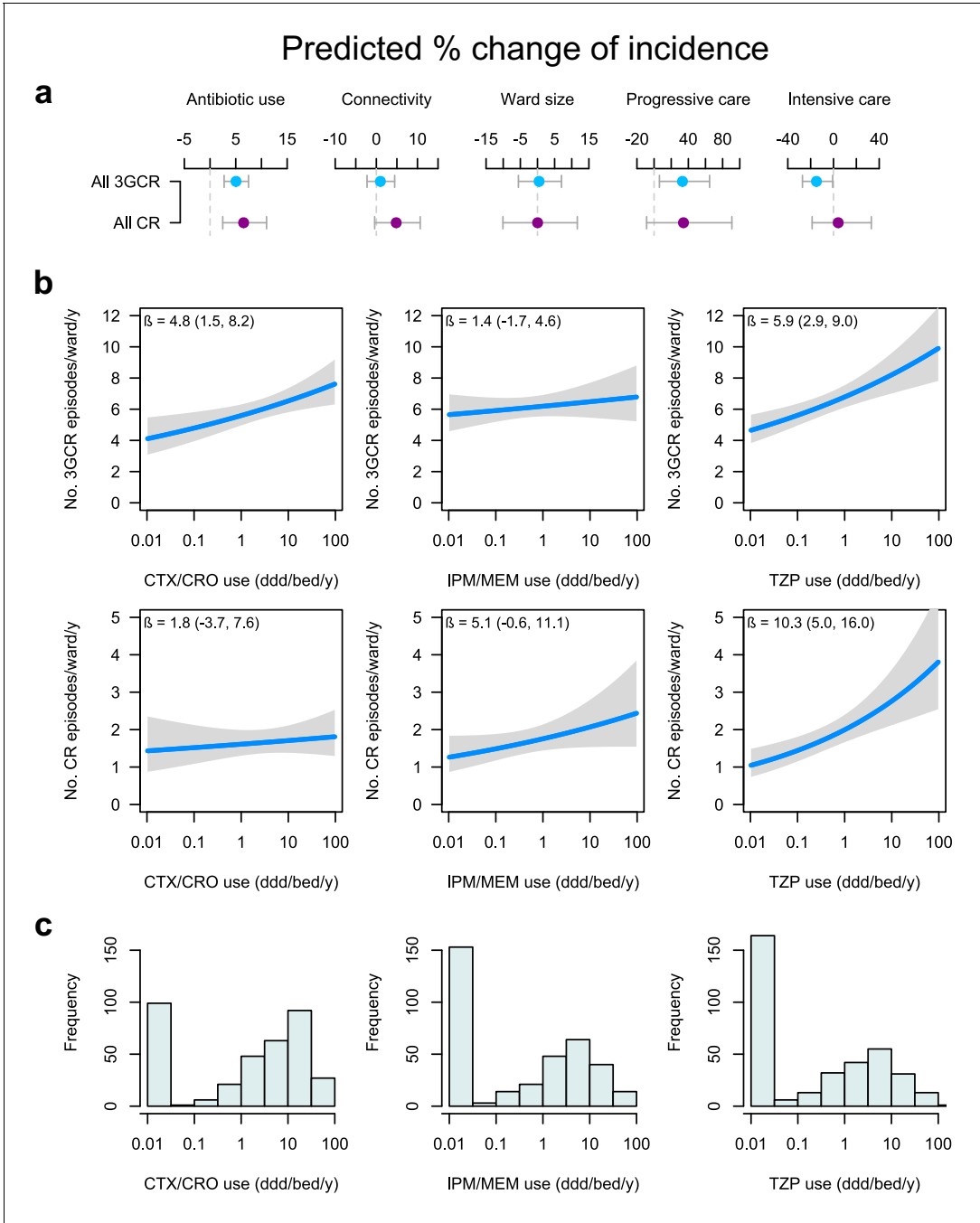

**Figure 3.** Global and specific antibiotics consumption predict the incidence of infection with 3rd-generation cepalosporin- or carbapenem-resistant ESKAPE2 pathogen variants. (**a**) Predicted percent change in incidence (points) with the 95% confidence interval (bars) of all 3GCR and CR infections for every doubling of antibiotic use, connectivity (estimated no. of patients infected with the same variant entering the ward), ward size (no. of beds); and ward type, based on quasi-Poisson regression models of the pooled incidence of 3GCR and CR infections in each ward (n = 357) that included the incidence control covariate (see Materials and methods). (**b**) Predicted incidence and 95% confidence bands of infections with 3GCR and CR pathogen variants depending on the consumption of CTX/CRO, IPM/MEM, and TZP, in models that included connectivity, the incidence control, and the consumption of 8 other antibiotic groups as covariates. (**c**) Consumption patterns of CTX/CRO, IPM/MEM, and TZP per ward in the hospital network. Variants classified as 3GCR were 3GCREC, 3GCRKP, CRKP, 3GCREB, CREB, PA, CRPA, AB, CRAB, EF, VREF, and MRSA; the CR category included CREC, CRKP, CREB, CRPA, CRAB, EF, VREF, and MRSA. Acronyms of pathogen variants and antibiotics are listed in *Tables 1* and *2*, respectively.

progressive care units and negatively associated with intensive care and blood cancer units (*Figure 3a*). These findings provide an unambiguous link between population-level antibiotic use and global resistance to empirical sepsis therapy that was robust to confounding by connectivity, microbiological sampling patterns, and other ward characteristics.

To identify the antibiotics whose use was most strongly associated with global carbapenem and cefotaxime-ceftriaxone resistance, we examined the effect of replacing the total antibiotic use in our models with individual antibiotic classes, similar to *Figure 2* models. Besides the consumption of cefotaxime-ceftriaxone and carbapenems, the consumption of piperacillin-tazobactam was a strong predictor of the incidence of both 3GC- or carbapenem-resistant infections (*Table 3*). To visualize the independent associations of incidence with the use of cefotaxime-ceftriaxone, carbapenems and piperacillin-tazobactam, we plotted the average ward-level infection incidence predicted by variations in the consumption volumes in models including the connectivity and incidence control covariates (*Figure 3b*). The incidence of 3GC-resistant infections was predicted by the consumption of cefotaxime-ceftriaxone (4.8% increase, 95% CI, 1.5 to 8.2%) but not carbapenems (1.4% increase, 95% CI, −1.7 to 4.6%). In the same vein, the use of carbapenems, but not 3GCs, predicted the incidence of carbapenem-resistant infections, although with a wide uncertainty margin (5.1% increase, 95% CI, −0.6 to 11.1%). Strikingly, the use of piperacillin-tazobactam predicted the incidence of both 3GC-resistant (5.9% increase, 95% CI, 2.9 to 9.0%) and carbapenem-resistant infections (10.3% increase, 95% CI, 5.0 to 16.0%). In both models, the amplitude of the piperacillin-tazobactam coefficient outweighed those of all other antibiotics (*Table 3*). Overall, these results indicate a specific association of cefotaxime-ceftriaxone and, to a slightly lesser extent, carbapenem use with resistance to the same antibiotic group, but not other groups, and identify a major role of piperacillin-tazobactam consumption in predicting the incidence of both 3GC- and carbapenem-resistant infections. To propose a unified ranking of the population-level impact of antibiotics on 3GC and carbapenem resistance, a final model was constructed by pooling all 3GC- and carbapenem-resistant variants together (*Table 3*). In this model, piperacillin-tazobactam and cefotaxime-ceftriaxone had the largest positive coefficients while 1GC/2GC and coamoxiclav had the largest negative coefficients (*Table 3*).

## Discussion

Understanding the respective impacts of antibiotic use and connectivity on the incidence of drug-resistant infections is essential for optimizing interventions against AMR. Pathogens whose incidence is strongly predicted by antibiotic use might be most effectively targeted by antibiotic restrictions.

**Table 3.** Associations between the consumption volume of 11 antibiotics and the cumulative incidence of 3GC- and/or carbapenem-resistant infections in 357 wards.

| | Predicted percent change in incidence (95% CI) per doubling of consumption volume | | |
|---|---|---|---|
| Antibiotics | 3GCR incidence model | CR incidence model | 3GCR or CR incidence model |
| TZP | 5.9 (2.9, 9.0) | 10.3 (5.0, 16.0) | 6.1 (3.1, 9.2) |
| CTX/CRO | 4.8 (1.5, 8.2) | 1.8 (-3.7, 7.6) | 4.6 (1.4, 8.0) |
| AMX | 2.7 (-1.5, 7.0) | 3.6 (-3.4, 11.2) | 2.7 (-1.4, 7.0) |
| CTZ/FEP | 1.7 (-1.0, 4.5) | 3.3 (-1.5, 8.3) | 1.6 (-1.1, 4.4) |
| IPM/MEM | 1.4 (-1.8, 4.7) | 5.1 (-0.6, 11.1) | 1.5 (-1.6, 4.8) |
| FQ | 0.5 (-1.9, 2.9) | −1.2 (-5.2, 3.0) | 0.4 (-1.9, 2.8) |
| OXA | −0.6 (-2.2, 1.1) | −2.0 (-4.6, 0.7) | −0.6 (-2.2, 1.0) |
| AMIN | −1.7 (-4.3, 0.9) | −1.1 (-5.4, 3.6) | −1.7 (-4.3, 0.9) |
| VAN/TEC | −2.3 (-4.9, 0.4) | −3.8 (-8.3, 0.9) | −2.3 (-4.9, 0.5) |
| 1GC/2GC | −2.1 (-4.0,−0.2) | −2.3 (-5.4, 1.0) | −2.1 (-4.0,−0.3) |
| AMC | −4.1 (-7.0,−1.1) | −5.8 (-10.6,−0.7) | −4.1 (-7.0,−1.1) |

NOTE. [a] Percent change was estimated from multivariable quasi-Poisson regression models that included the connectivity and the incidence control covariates (see Materials and methods). 3GCR, 3rd-generation cephalosporins-resistant infections; CR, carbapenem-resistant infections. Acronyms of antibiotics are listed in *Table 2*.

On the other hand, infection control interventions might be most relevant against pathogens whose incidence is predicted by connectivity. By applying a metapopulation framework to explain variations in infection incidences across a large network of hospital wards, we found that both antibiotic use and connectivity independently contribute to ward-level AMR in several pathogen species. Our study also provides the first quantitative ranking of the predicted impact of several key antibiotics on the global burden of drug-resistant infections in a hospital network.

While previous theoretical work based on modeling and simulation has predicted how patient transfers contribute to AMR prevalence through pathogen dissemination, they have typically considered only a single resistant variant or used a generic simulated pathogen (*Donker et al., 2017*; *Donker et al., 2010*; *Vilches et al., 2019*). Our study provides empirical evidence to support these predictions in general, while identifying substantial variation in the response to connectivity between different species and variants (*Figure 1*). In our closed network model of the hospital, the influence of connectivity on the incidence of a pathogen variant is expected to be higher if the variant is endemic to the hospital, its prevalence varies across wards, and changes from one variant category to another are rare. If a variant is frequently introduced from outside of the network, the contribution of within-network movements, hence of connectivity, to the prediction of the local incidence will be diminished. The measured influence of connectivity should also be reduced if a variant frequently undergoes transitions between resistance categories (e.g., by evolving or losing resistance) because local emergence will affect incidence more strongly than inter-ward introduction.

Consistent with this theoretical interpretation of connectivity, we found that its influence was strongest in the hospital-endemic pathogens *P. aeruginosa* and *E. faecium* (*Blanc et al., 2007*; *Wurster et al., 2016*; *Zhou et al., 2020*). Intriguingly, MRSA incidence was not predicted by connectivity. This is at odds with the classical perception of MRSA as typically nosocomial and, by extension, hospital-endemic. However, available evidence does not strongly support the qualification of MRSA as a hospital-endemic pathogen in our setting. In France, the proportions of MRSA among *S. aureus* are comparable in community and hospital settings, at about 10% (*ECDC, 2019*; *ONERBA France, 2018*; *Santé Publique France, 2019*). MRSA population structures in hospitals and the community are comparably dominated by the so-called ST8 Lyon clone which is equally found in in- and outpatients (*Dauwalder et al., 2008*). Finally, MRSA infections were especially diffuse in our network, with a concentration index even lower than that of the less-resistant *S. aureus* infections (*Table 1*). Collectively, this does not support the conclusion that MRSA concentrates in French hospitals compared to the community, which might explain why connectivity did not predict MRSA incidence in our study.

The incidence of global carbapenem-resistant infections was strongly predicted by connectivity (*Figure 3*), with an effect size comparable to that of antibiotic use, while connectivity did not predict global 3GC-resistant infections. This suggests that reducing connectivity with infection control interventions could be more effective at preventing the spread of carbapenem-resistant pathogens between wards compared to 3GC-resistant pathogens. Contrasting with hospital-endemic pathogens, connectivity was a comparatively weaker predictor for community-associated variants (3GC-resistant *E. coli* and *K. pneumoniae*) that enter from outside the hospital network, and in variants whose resistance is selected locally because they can frequently shift between susceptible and resistant categories. Indeed, the weak association of connectivity with the incidence of resistant *E. cloacae* complex variants, for instance, might be explained by the plasticity of their resistance profile, facilitating their local selection. While resistance in *E. coli* and *K. pneumoniae* typically requires gene acquisition (*Manges et al., 2019*; *Wyres and Holt, 2016*), *E. cloacae* complex can resist cephalosporins and carbapenems through increased AmpC beta-lactamase and decreased porin expression (*Pavez et al., 2016*; *Babouee Flury et al., 2016*; *Lee et al., 2017*). Such resistance emerges through adaptation and de novo mutations that are rapidly selected from the local reservoir of susceptible progenitors under antibiotic selection pressure (*Hawken et al., 2018*; *Moradigaravand et al., 2016*).

Ecological studies have repeatedly identified associations between the use of antibiotics and AMR prevalence, but such associations do not necessarily reflect AMR selection (*Schechner et al., 2013*). Specific associations at the level of antibiotic and variant pairs are not all equally likely to result from selection by antibiotics or from increased antibiotic use in response to AMR. Based on medical and biological reasoning, we identified associations more likely to represent possible selection and showed that the average strength of these associations outweighed the others (*Figure 2*).

Noteworthy, several associations with a sizeable strength could not be formally classified as possibly selective but might reflect co-selection. The association of cefotaxime-ceftriaxone with carbapenem-resistant variants in *K. pneumoniae*, but not in *E. coli*, possibly reflected selection for the highly frequent 3GC-resistance in carbapenemase-producing *K. pneumoniae*. This contrasts with *E. coli* $Oxa_{48}$ producers that frequently remain susceptible to 3GCs but resistant to piperacillin-tazobactam (*Huang et al., 2014*), consistent with the strong association between the use of piperacillin-tazobactam and the incidence of carbapenem-resistant *E. coli* infections (*Figure 2a*). These findings suggest that the associations between antibiotic use and resistance preferentially reflected AMR selection by antibiotics rather than adaptation of antibiotic use in response to AMR. However, this observation does not eliminate temporal ambiguity between antibiotic use and resistance in our models. Further research using, for instance, multi-state models or time-series analyses, may better clarify this ambiguity.

Because intrinsic and acquired resistances to an antibiotic equally lead to treatment failure, we modeled the pooled incidences of infections with 3GC- or carbapenem-resistant variants of the $ESKAPE_2$ pathogens, including those with intrinsic resistance. This approach allowed us to rank antibiotics by their potential association strength with global AMR. The use of piperacillin-tazobactam and cefotaxime-ceftriaxone predicted 3GC resistance and the use of piperacillin-tazobactam predicted carbapenem resistance (*Figure 3*). The positive association of the use of piperacillin-tazobactam with both 3GC- and carbapenem-resistant infections deserves further attention. Based on its in vitro efficacy against extended-spectrum beta-lactamase (ESBL)-producing enterobacteria, piperacillin-tazobactam has been repeatedly considered as an alternative drug of choice in carbapenem-sparing strategies (*Harris et al., 2015*; *Peterson, 2008*). The strategy of replacing carbapenems with piperacillin-tazobactam assumes: (1), that piperacillin-tazobactam is clinically as effective as carbapenems on piperacillin-tazobactam-susceptible pathogens; and (2), that AMR selection under piperacillin-tazobactam pressure is weaker than under carbapenem pressure, as reflected by a recent consensus-based ranking of beta-lactams for de-escalation therapy (*Weiss et al., 2015*). Yet, in several recent reports including a multicenter randomized clinical trial, piperacillin-tazobactam treatment of sepsis with ESBL-producing enterobacteria was associated with poorer outcomes compared to carbapenem treatment (*Harris et al., 2018*; *Ofer-Friedman et al., 2015*; *Tamma et al., 2015*). From an epidemiological standpoint, studies of the respective associations of piperacillin-tazobactam and carbapenem use with AMR yielded conflicting results. At the population level, the incidence of carbapenem-resistant enterobacteria was negatively associated with the use of piperacillin-tazobactam in a 5 year, single-hospital trend analysis study (*McLaughlin et al., 2013*). At the patient level, however, the exposure to piperacillin-tazobactam was associated with the acquisition of carbapenem-resistant *P. aeruginosa* in a meta-analysis (*Raman et al., 2018*) and carbapenem-resistant Gram-negative bacilli in a single-hospital prospective cohort study (*Marchenay et al., 2015*). If confirmed in other settings, our finding that piperacillin-tazobactam use correlates with both 3GC- and carbapenem-resistant infections might call for a reevaluation of the rationale of recommending piperacillin-tazobactam over other drugs for ecological reasons. As a note of caution, our observation that piperacillin-tazobactam use predicted a higher incidence of 3GC- and carbapenem-resistant variants does not imply selection for acquired resistance through any specific mechanism such as carbapenemase production. We also note that the link between piperacillin-tazobactam use and global resistance resulted from the accumulation of small, positive associations with most 3GC- and carbapenem-resistant variants, including those with intrinsic resistance (*Figure 2*). Because of this, links between piperacillin-tazobactam use and 3GC- or carbapenem-resistant infections might go undetected in studies focusing on individual pathogen variants.

Our study has several limitations. First, we did not consider the role of healthcare workers in pathogen transmission, nor the role of direct patient admissions from the community. Second, although our 3-level categorical coding of ward types captures variations in patient fragility within our study system, such a coarse-grained classification could potentially leave residual confounding. Third, the small sample sizes of resistant variants of *E. faecium* and *A. baumannii* limited our ability to draw robust inferences regarding these variants and further studies are required to confirm these results (*Arias and Murray, 2012*; *Hsu et al., 2017*). Finally, our findings reflect the AMR ecology of a Western European area, with generally lower prevalences of carbapenemase-producing pathogens, vancomycin-resistant *E. faecium*, and MRSA than in other regions of the world.

To conclude, the modeling of the incidence of infections with seven major bacterial species and their drug-resistant variants in hospital wards using a metapopulation framework indicates that both antibiotic use and inter-ward connectivity may predict the burden of AMR in a variant-specific fashion. This supports the need to tailor strategies against AMR to the targeted pathogen. Along with novel hospital-level insights into the drivers of AMR, our work illustrates the application of the methodological framework of metapopulation ecology to the problem of hospital AMR. Applying this framework to other healthcare settings could help inform the local and regional antibiotic stewardship and infection control strategies.

## Materials and methods

### Data collection and compilation

We obtained data on infection incidence from the information system of the Institut des Agents Infectieux, the clinical microbiology laboratory of the Hospices Civils de Lyon, a group of university hospitals serving the Greater Lyon urban area (~1.4 million inhabitants) of France. For each ward from October 1$^{st}$, 2016 to September 30$^{th}$, 2017, we extracted the number of clinical samples, after exclusion of screening samples, positive for at least one of the ESKAPE$_2$ species (as determined using Vitek MS MALDI-ToF identification, bioMérieux), falling into one of the resistance variant categories defined in *Table 1*. Resistance was based on available results for susceptibility to, where applicable, ceftriaxone, cefotaxime, ceftazidime, cefepime, imipenem, meropenem, oxacillin, and vancomycin. Samples were deduplicated per patient, ward, and pathogen variant, so that only the first sample positive for the same variant in the same ward was considered for each patient. Patients positive for multiple variants and/or sampled from multiple wards were considered as multiple, distinct infection episodes. Antibiotic use in defined daily doses (ddd) of all systemic antibacterial drugs (ATC classification term J01), as well as of specific (groups of) molecules defined in *Table 2*, were extracted from the pharmacy department information system. For each pair of wards, the number of patient transfers was extracted from the hospital information system along with, for each ward, the number of beds, the type of medical activity and the number of patient admissions. Because of the aggregated nature of the data, informed consent was not sought, in accordance with French regulations. Our main response variable was the number of patients per ward infected with each pathogen variant, expressed as incidence over 1y.

### Controlling for microbiological sampling patterns

The estimation of a causal effect between antibiotic use and the incidence of infections with a given pathogen variant is biased by variations in the distribution of infections across wards, because this distribution influences both antibiotic prescriptions and microbiological sampling efforts. A causal network representation of this situation is shown in *Figure 1—figure supplement 2*. The distribution of infections is difficult to determine and to represent as a model covariate, preventing its direct inclusion in our models. To circumvent this issue, the unrepresentable distribution of infections was replaced with a proxy variable that we called the incidence control, designed to capture variations of microbiological sampling frequencies and of the sampled anatomic sites (e.g. urinary vs. respiratory tract) across wards. Of note, ward size and type were also assumed to correlate with the infection distribution and to contribute to the proxy adjustment. However, the inclusion of ward size and type had a negligible impact on model fit compared to the incidence control value, suggesting that this latter variable captures most of the signal.

The incidence control was defined as the expected incidence of a pathogen variant explained by microbiological sampling alone, under the assumption that the pathogen incidence is conditionally independent of the ward given the sampled anatomic site. Anatomic sites were assigned to seven site groups, namely, skin and soft tissues, respiratory tract, urinary tract, digestive tract, vascular access devices, sterile sites (such as cerebrospinal fluid and peripheral blood cultures) and other sites.

The incidence control was computed as follows for each pathogen variant in each ward. In this section, wards are indexed by $i = 1, \ldots, 357$, groups of anatomical sites are indexed by $j = 1, \ldots, 7$, and pathogen variants are indexed by $k = 1, \ldots, 17$. For conciseness, we introduce the special case $k = 0$ to denote a sample not positive for any of the considered pathogen variants. Contrary to the

incidence calculation described in the previous section, all samples were considered without deduplication to account for how repeated sampling increases the likelihood of pathogen detection.

First, for each pathogen variant $k$ and group of anatomic sites $j$, we computed the average probability of variant detection aggregated for all wards in the network as

$$P(Variant = k | Site = j) = \frac{\sum_{i=1}^{357} N(Variant = k, \, Site = j, \, Ward = i)}{\sum_{i=1}^{357} N(Ward = i, \, Site = j)}$$

where $N(Variant = k, \, Site = j, \, Ward = i)$ is the number of samples in ward $i$ taken from site $j$ positive for the variant $k$ (or, if $k = 0$, negative for all variants) and the denominator denotes all samples taken irrespective of their result. For simplicity, we assumed independence between the pathogen variants isolated from a same sample and between samples collected from a same patient. Of note, most patients (n = 22,646 / 26,064; 86.9%) were sampled only once.

To account for repeated sampling, we considered the probability that $M$ samples from a site $j$ in a patient remained negative for pathogen variant $k$,

$$P(Variant \neq k | M, Site = j) = \left[1 - P(Variant = k | Site = j)\right]^{M}.$$

Sampling from multiple sites was represented using the vector notation $\boldsymbol{M} = \{M_1, \ldots, M_7\}$ to denote the respective numbers of samples taken from site groups 1 to 7. For instance, $\boldsymbol{M} = \{1, 0, 2, 0, 0, 0, 0\}$ denotes 1 skin or soft tissue sample and 2 urinary samples. Using this notation, the probability that all samples from all sites remained negative is,

$$P(Variant \neq k | \boldsymbol{M}) = \prod_{j=1}^{7} P(Variant \neq k | M_j, Site = j),$$

and the probability that at least one sample was positive for variant $k$ is,

$$P(Variant = k | \boldsymbol{M}) = 1 - P(Variant \neq k | \boldsymbol{M}).$$

This relationship was used to calculate the expected incidence of variant $k$, that is the number of patients with at least one sample positive for $k$, by considering in each ward $i$ the number $N(i, \boldsymbol{M})$ of patients with the same number of samples from each site $\boldsymbol{M}$ and the probability of being tested positive given $\boldsymbol{M}$. The expected incidence of variant $k$ in ward $i$ was then defined as,

$$N(Variant = k | Ward = i) = \sum_{\boldsymbol{M} \in \Omega} N(i, \boldsymbol{M}) \times P(Variant = k | \boldsymbol{M})$$

where $\Omega$ denotes the set of possible sample combinations.

Clearly, variations in the incidence control value between wards depend only on the number and sites of microbiological samples taken, thus reflecting the incidence and types of bacterial infections at ward-level independent of between-ward variations of pathogen community structure. Under our assumption that the incidence control is a valid proxy to the unrepresentable distribution of infections in each ward, the incidence control should correlate both with antibiotic use and the incidence of infections. Bivariate analyses confirmed that the incidence control correlated with the observed cumulative incidence of all bacteria ($R^2$=0.96, 95% CI, 0.95 to 0.96, *Figure 1—figure supplement 3*) and, to a lesser extent, with the total antibiotic use ($R^2$=0.34, 95% CI, 0.25 to 0.40, *Figure 1—figure supplement 4*). These correlations remained substantial for most pathogen variants and specific antibiotics. The incidence control was added as an adjustment covariate in all models predicting infection incidence. The adjusted models, thus, predicted the incidence of infections in excess of what would be expected based on variations in sampling intensity alone.

## Connectivity and other ward characteristics

In ecology, habitat quality refers to the resources and conditions that allow individuals and populations to persist in a location (*Hall et al., 1997*), such as food or cover from predators (*Johnson, 2005*). We described the habitat quality of hospital wards for bacterial pathogens using

explanatory variables adapted from Hanski's metapopulation models (*Hanski, 1998*; *Hanski, 1994*), namely patch size and connectivity, along with additional variables capturing patient fragility and antibiotic selection pressure. We considered each ward within the hospital system as a distinct habitat patch, *i*. We used the number of beds both as a measure of patch size (capturing the number of patients available for colonization) and as a proxy for contact opportunities between patients within the same ward.

The connectivity estimate was implemented as a proxy to the unobservable number of introductions of each variant in each ward during the study period (*Donker et al., 2017*). To estimate this quantity, we measured directional, partial connectivity $S_{j,i}$ from ward *j* to ward *i* as the number of patients transferred directly from *j* to *i,* times the inferred probability that each patient tested positive for the variant. Partial connectivity, thus, was the expected number of positive patients transferred from *j* to *i*. Finally, connectivity for ward *i* was the sum of all directional connectivities, $S_i = \sum_j S_{j,i}$. This estimation procedure relies on simplifying assumptions, namely that all positive cases are detected and remain positive upon transfer to the downstream ward; and that the probability and destination ward of a patient transfer does not depend on infection.

Along with size and connectivity, wards were characterized by ward type based on patient fragility, and antibiotic consumption. Ward type was coded as a categorical variable with the following three levels: general wards, intermediate (progressive) care units, and intensive care and blood cancer units. Antibiotic use was normalized by dividing by the number of beds in each ward and expressed in ddd/bed/y.

## Statistical analysis

The statistical unit was the individual ward (n = 357) in all analyses. We used the asymptotic Simpson index (*Simpson, 1949*), also known as the Hunter-Gaston index (*Hunter and Gaston, 1988*), to determine the probability that two random isolates of a given variant were isolated in the same ward or that two random doses of a given antibiotic were delivered in the same ward. The index is defined as $[\sum_i n_i (n_i\text{-}1)]/[N (N\text{-}1)]$ where $n_i$ is the number of infection episodes (or antibiotic doses) detected (or delivered) in ward *i* and $N_i = \sum_i n_i$ is the total number of infection episodes (or antibiotic doses). In ecology, the Simpson index is typically used to measure biodiversity by estimating the probability that two individuals from a sample belong to the same species (*Simpson, 1949*). Here, we use this index to examine the distribution of sampling locations relative to the taxa, measuring the probability that two infection episodes occur, or two antibiotic doses are consumed, in the same ward. Hence, we used the term 'concentration index' to avoid confusion with a diversity measure. The *iNext* R package was used to determine bootstrap-based 95% confidence intervals of the concentration index (*Hsieh et al., 2016*).

Models of infection incidence were constructed using multivariable quasi-Poisson regression of the form

$$\mathrm{E}(\textit{Incidence }|\boldsymbol{x}) = \exp(\textit{Intercept} + \beta_1 x_1 + \beta_2 x_2 + \ldots), \ \ \mathrm{Var}(\textit{Incidence}) = \theta\,\mathrm{E}(\textit{Incidence }|\boldsymbol{x})$$

where $\mathrm{E}(\cdot)$ denotes expectation, $\boldsymbol{x}$ is the vector of predictors, such as antibiotic consumptions, the $\beta$'s are model coefficients, $\mathrm{Var}(\cdot)$ denotes variance and $\theta$ is the overdispersion parameter of the quasi-Poisson model. The overdispersion parameter is used to relax the Poisson assumption of equality of the expectation and variance of the response variable. We used the quasi-Poisson distribution because we found evidence of both under- and overdispersion, as evidenced by fitted quasi-Poisson dispersion parameters ranging from 0.24 (strong underdispersion, found for carbapenem-resistant *A. baumannii* in *Figure 1* model) to 2.5 (moderate overdispersion, found for less-resistant *E. faecium*) (*Supplementary file 1* Table 1b). We favored the quasi-Poisson model over the alternative, negative binomial model because the latter gives greater weight to smaller sites, whereas the former gives greater weight to the larger sites (*Ver Hoef and Boveng, 2007*). In our application, greater weight should be given to wards with greater incidence rather than to those with less.

We constructed a model for each variant. The response variable was the deduplicated patient counts. All non-categorical explanatory variables including the incidence control, ward size, connectivity and antibiotic use were log$_2$-transformed before further analyses. To avoid negative infinity values from this transformation, all zeroes were first converted to half the minimum non-zero value. This transformation was associated with better model fit (using the model structure of *Figure 1*), in

terms of Akaike information criterion (with Poisson distribution), compared with: (1) replacing zeroes with the minimum non-zero value before taking logs; (2) adding one to zero values before taking logs; or (3) avoiding log transformation. The $\log_2$ transformed data were used for all subsequent analyses.

To ease interpretation of model coefficients, we converted the raw coefficients $\beta$ from all models to percent changes in infection incidence, equal to $100 \times \left(e^{\beta} - 1\right)$. All analyses used R software version 3.6.0.

## Possibly selective associations between antibiotic use and resistance

We examined criteria to identify a priori possibly selective associations between antibiotic use and resistance. The criteria were based on medical and biological considerations, namely, that antibiotics inactive against a variant are unlikely to be prescribed in response to this variant's prevalence; and that antibiotics are most likely to select for a variant when resistance provides a specific advantage, hence, when the variant is not resistant to more potent antibiotics (e.g., CTX/CRO is more likely to select for PA than for CRPA in which carbapenem resistance provides no additional benefit under CTX/CRO pressure). This rationale led to the following criteria: (1) the variant is always resistant to the antibiotics of interest; (2) the variant is not resistant to antibiotics more potent (in terms of spectrum or efficacy) than the antibiotics of interest; and (3) the antibiotics of interest can be plausibly used against the variant in empirical therapy. A total of 19 associations fulfilled the criteria for possibly selective associations: 3GCREC with CTX/CRO; CREC with IPM/MEM; KP with AMX; 3GCRKP with CTX/CRO; CRKP with IPM/MEM; EB with AMC; C3GREB with CTX/CRO and TZP; CREB with IPM/MEM; PA with CTX/CRO; CRPA with IPM/MEM; SA with AMX; MRSA with C1G/C2G, OXA and AMC; AB with CTX/CRO; CRAB with IPM/MEM; EF with CTX/CRO; and VREF with VAN/TEC. Of note, the associations not fulfilling the criteria for possible selection can be interpreted as negative controls in our models, and their coefficients are expected to be distributed around zero (null distribution) in the absence of residual confounding. In line with this interpretation, the coefficients of most of the negative control associations followed a near-zero-centered distribution (*Figure 2c*), suggesting that residual confounding was negligible in the adjusted models.

## Pooled analysis of CTX/CRO- and IPM/MEM-resistant variants

To model the cumulative incidences of 3GC- and carbapenem-resistant infections, pathogen variants were pooled into resistance categories. When resistance to CTX/CRO or IPM/MEM was not determined by design (such as 3GC resistance in 3GCREC) or by intrinsic resistance (such as 3GC resistance in *E. faecium*), variants were classified as resistant when the proportion of resistance in our setting was above 80%. This less-stringent resistance criterion, compared to the criterion used to determine possibly selective associations, was chosen to avoid the exclusion of variants that are mostly resistant to an antibiotic group, which would bias pooled analyses. Applying the 80% threshold for the proportion of resistance led to classifying CRKP and CREB as 3GC-resistant (91% and 93% 3GC resistance, respectively) but not CREC (61% 3GC resistance); and EF and VREF as carbapenem-resistant (84 and 100% carbapenem resistance, respectively, inferred from ampicillin resistance [*Weinstein, 2001*]). Overall, the 3GCR category included 3GCREC, 3GCRKP, CRKP, 3GCREB, CREB, PA, CRPA, AB, CRAB, EF, VREF, and MRSA; and the CR category included CREC, CRKP, CREB, CRPA, CRAB, EF, VREF, and MRSA.

## Data and software code availability

All data and software code that support the findings of this study are available at: *Shapiro, 2020*, https://github.com/rasigadelab/metapop (copy archived at https://github.com/elifesciences-publications/metapop).

## Acknowledgements

The authors thank Arnaud Friggeri, Alain Lepape, and Florent Wallet for fruitful discussion. Julie Teresa Shapiro, Gérard Lina, François Vandenesch, and Jean-Philippe Rasigade are part of the French Laboratory of Excellence project ECOFECT (ANR-11-LABX-0048).

## Additional information

### Funding

| Funder | Grant reference number | Author |
|---|---|---|
| Fondation Innovations en In-fectiologie | R18037CC | Jean-Philippe Rasigade |
| French Laboratory of Excel-lence project ECOFECT | ANR-11-LABX-0048 | Julie Teresa Shapiro<br>Gilles Leboucher<br>François Vandenesch<br>Jean-Philippe Rasigade |

The funders had no role in study design, data collection and interpretation, or the decision to submit the work for publication.

### Author contributions

Julie Teresa Shapiro, Conceptualization, Software, Formal analysis, Visualization, Methodology, Writing - original draft, Writing - review and editing; Gilles Leboucher, Anne-Florence Myard-Dury, Pascale Girardo, Anatole Luzzati, Sandrine Couray-Targe, Data curation, Investigation, Writing - review and editing; Mélissa Mary, Jean-François Sauzon, Data curation, Formal analysis, Investigation, Writing - review and editing; Bénédicte Lafay, Validation, Writing - review and editing; Olivier Dauwalder, Investigation, Writing - review and editing; Frédéric Laurent, Christian Chidiac, François Vandenesch, Writing - review and editing; Gerard Lina, Conceptualization, Writing - review and editing; Jean-Pierre Flandrois, Conceptualization, Investigation, Writing - review and editing; Jean-Philippe Rasigade, Conceptualization, Software, Formal analysis, Supervision, Funding acquisition, Investigation, Methodology, Writing - original draft, Project administration, Writing - review and editing

### Author ORCIDs

Julie Teresa Shapiro (ID) https://orcid.org/0000-0002-4539-650X
Gilles Leboucher (ID) https://orcid.org/0000-0001-7043-9834
Bénédicte Lafay (ID) https://orcid.org/0000-0001-5783-5269
Olivier Dauwalder (ID) https://orcid.org/0000-0003-1722-1582
Christian Chidiac (ID) https://orcid.org/0000-0001-7002-2713
Jean-Pierre Flandrois (ID) https://orcid.org/0000-0002-4953-9125
Jean-Philippe Rasigade (ID) https://orcid.org/0000-0002-8264-0452

### Decision letter and Author response

Decision letter https://doi.org/10.7554/eLife.54795.sa1
Author response https://doi.org/10.7554/eLife.54795.sa2

## Additional files

### Supplementary files

• Supplementary file 1. Supplementary tables. (**a**) Summary statistics (mean, interquartile range) of connectivity for each variant. (**b**) Comparison of log-likelihood values from multivariable Poisson and negative binomial regressions and overdispersion parameters from quasi-Poisson multivariable regression of the incidence of infections with 17 pathogen variants. All models included total antibiotic consumption, connectivity, ward size, ward type, and the incidence control value as variables.

• Transparent reporting form

### Data availability

All data and software code that support the findings of this study are available at: https://github.com/rasigadelab/metapop (copy archived at https://github.com/elifesciences-publications/metapop).

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
