## [Decision Letter]

**Acceptance summary:**

A metapopulation is a group geographically separated sub-populations of a species where there are interactions between the sub-populations. The concept has been usefully applied to understand the impact of patient movements on the spread of drug-resistant bacteria in healthcare settings on a number of previous occasions, but this paper represents the first application of the concept to a large data-set including multiple bacterial species and detailed data on antibiotic usage, another key driver of the spread of resistance. The analysis provides important new insights about how patterns of patient movement within healthcare settings and antibiotic exposures both contribute to the spread of antimicrobial resistance.

**Decision letter after peer review:**

Thank you for submitting your article "Metapopulation ecology links antibiotic resistance, consumption and patient transfers in a network of hospital wards" for consideration by *eLife*. Your article has been reviewed by two peer reviewers, and the evaluation has been overseen by a Reviewing Editor and Eduardo Franco as the Senior Editor. The following individuals involved in review of your submission have agreed to reveal their identity: Tjibbe Donker (Reviewer #1); Rene Niehus (Reviewer #2).

The reviewers have discussed the reviews with one another and the Reviewing Editor has drafted this decision to help you prepare a revised submission.

As the editors have judged that your manuscript is of interest, but as described below that additional analysis may be required before it is published, we would like to draw your attention to changes in our revision policy that we have made in response to COVID-19 (https://elifesciences.org/articles/57162). First, because of disruptions to normal working life for many, we will give authors as much time as they need to submit revised manuscripts. We are also offering, if you choose, to post the manuscript to bioRxiv (if it is not already there) along with this decision letter and a formal designation that the manuscript is 'in revision at *eLife*'. Please let us know if you would like to pursue this option. (If your work is more suitable for medRxiv, you will need to post the preprint yourself, as the mechanisms for us to do so are still in development.)

Summary:

In this work, the authors study the very important question of how population level antibiotic resistance in clinically important bacterial taxa is driven by ward-level antibiotic use and patient transfer between wards. For this the authors use clinical specimen data, antibiotic use data and ward-connectivity data aggregated across one year. They perform a between ward comparison by applying ecological metapopulation methods to correct for confounding. They find, in agreement with previous evidence, that nosocomial and carbapenem resistance pathogen incidence is positively associated with connectivity, and that antibiotics have varying effects depending on the pathogen variant, with Piperacillin-tazobactam appearing to select most strongly for pathogens resistant to empirical treatment. More broadly, this would mean that different pathogen variants require either antibiotic stewardship or patient isolation interventions.

Essential revisions:

Currently the description of the methods is incomplete and makes it difficult to assess the validity of the results. Specifics are described below.

1) Is this incidence control value (Incidence(Variant|Ward) ) calculated for each of the 357 wards, or for each of the 3 ward types? This should be made clearer in the text, and also by introducing indexing into the mathematical expressions.

2) The authors build a method to account for

i) variants being associated with diff anatomical sites P(Variant|Location)

ii) different wards differing in which anatomical sites are typically sampled, and in how many samples are typically taken per patient P( No Variant | N, Ward )

From the definition of P( No Variant | N, Ward ) it is not clear whether the authors also account for the fact that samples from one patient are not independent (in terms of the sampled anatomical site) and a definition of N(I,Ward) is missing. Is this the number of samples specific from a patient with index i, being disaggregated into location and ward, or is it the total number of samples in the ward disaggregated into the different anatomical sites? The difference is important: if N(I,ward) is the total number of ward samples disaggregated into anatomical sites, then the expression P(No Variant|N, Ward) assumes that samples taken from a patient i are independent.

3) In Data collection and compilation the authors say that clinical samples are deduplicated, resulting in one sample per patient, but in the section after it seems that multiple samples per patient are used for to compute incidence control. It is correct that for the incidence control value samples where used without deduplication, but for the regression model fit were deduplicated counts used? This should be made clearer.

4) Using a priori expected selective antibiotic-variant pairs the authors show that the model manages to pick up more of those pairs than others. This is expected: even when substantial confounding increases all individual antibiotic-variant associations, a direct causal effect in a abx_A_-variant_X_ pair is expected to make it easier to pick up this association. It is not clear from the Materials and methods whether a multivariate model was used here. Please provide the regression equation in the Materials and methods to allow the reader to follow what was done. If it is univariate, it is to be expected that looking into different individual antibiotics will introduce a lot of new confounding from association with other antibiotic treatments. What would be useful here is a multivariate analysis of individual antibiotics (asking about the effect of each, given the effect of all the other treatment), and importantly including a few additional predictors (treatments) that one would expect to NOT be correlated with AMR (negative control). A zero-centred distribution of the negative control β parameters would then indicate lack of confounding.

5) A Poisson model was used. Typically Negative Binomial models will fit these type of data better as there's usually some overdispersion. The effect will be wider confidence intervals. It would be helpful to compare fits of Poisson and Neg Bin and go with the best fitting model.

6) The repeated dichotomising between "significant" and "non-significant findings" in the paper is at odds with current statistical thinking (see, for example, the American Statistical Association consensus statement on p-values https://www.amstat.org/asa/files/pdfs/P-ValueStatement.pdf). We strongly suggest the authors drop this terminology and instead of reporting results as significant or not, report point estimates and confidence intervals for everything.

7) The exclusion of rarely used antibiotics from the analysis makes sense, but it wasn't clear why narrow spectrum antibiotics or those used in combination were also excluded. We think including these in the analysis would improve the paper.

8) There was a consensus that the conclusions were too strong e.g. "Our data establish that.…". As these are observational data, with lots of unmeasured confounding likely, the conclusions should better reflect the uncertainty.

9) "Hospital-based antibiotic stewardship is a cornerstone strategy to combat AMR, based on the assumption that AMR evolves in hospitals. Yet, there is surprisingly limited evidence to support this assumption^47-49^"

It's not completely clear what this statement means (there are lots of examples where antibiotic use in hospitalised patients clearly selects for resistant variants e.g. fusidic acid resistance in staphylococci), but if it's referring to population wide selection mediated by antibiotic use there are several additional papers that could be cited ( eg several papers by José-María López Lozano and Timothy Lawes, such as their Nature Micriobiology 2019 paper or more recently Tom Crellen's 2019 paper on *Klebsiella* in *eLife*).

[Editors' note: further revisions were suggested prior to acceptance, as described below.]

Thank you for submitting your article "Metapopulation ecology links antibiotic resistance, consumption, and patient transfers in a network of hospital wards" for consideration by *eLife*. Your article has been reviewed by two peer reviewers, and the evaluation has been overseen by a Reviewing Editor and a Senior Editor. The following individuals involved in review of your submission have agreed to reveal their identity: Tjibbe Donker (Reviewer #1); Ben S Cooper (Reviewer #3).

The reviewers have discussed the reviews with one another and the Reviewing Editor has drafted this decision to help you prepare a revised submission.

The manuscript is much improved both methodologically and in terms of readability, and key points raised in the first round of reviews have been addressed. There are, however some remaining concerns which are important to address.

Essential revisions:

1) In many places the paper refers to multivariate (or bivariate) models. Though this terminology is widely misused, multivariate (still) means models with multiple dependent variables. Models with multiple independent variables but a single dependent variable are *multivariable* models. The models used here seem to be multivariable and not multivariate.

2) In most places the authors have rightly been cautious about reaching causal conclusions based on the associations found in the observational data. However there are still a small number of places where results are presented using causal language (i.e. language that many readers are likely to interpret as implying a direct causal relationship) that is not fully justified e.g "with every doubling of antibiotic use increasing incidence by 49%", "connectivity had a larger effect on the incidence". The manuscript would be improved by rephrasing results to report these as associations, and the discussion can then consider the plausibility of a causal interpretation (as has already been done in most cases).

3) "This nearly zero-centered distribution of coefficients in associations not expected to represent direct selection suggests that residual confounding in the models was negligible….": We would suggest revising this sentence to make it easier to understand.

4) "If the resistant infections in a ward mostly result from the admission of already colonized patients, antibiotic restrictions may have a limited impact on AMR compared to infection control measures that prevent the further dissemination of the pathogens" This makes sense if "limited impact on AMR" refers to colonisation with AMR (though it's not clear that it does given that the paper is concerned with clinical infection rather than carriage data), but antibiotics can also act to select for resistance within hosts increasing (or reducing) the chance of clinical infection with resistant organisms (see, for example, Niehus et al., 2020 and references therein). We strongly encourage the authors to rephrase to acknowledge that antibiotics can also have a big impact on within host dynamics and that antibiotic restrictions (or other changes) can also have an important impact even if most infections occur in already colonised patients.

5) "the influence of connectivity on the incidence of a pathogen variant is expected to be higher if the variant is endemic to the hospital". This seems intuitive but if the level of endemicity is so high that all patients are colonised (as we would expect for E coli) then surely we would expect connectivity to have no effect on the incidence of infection with the pathogen variant. So is this statement generally true?

6) "Consistent with this theoretical interpretation of connectivity, we found that its influence was strongest in the typical nosocomial pathogens *P. aeruginosa* and *E. faecium*." Though isn't it also the case that there was little evidence of an association between connectivity and MRSA infection incidence, but MRSA (or, at least, many lineages of MRSA) also represents a "typical nosocomial pathogen". It seems as important to highlight results that don't fit with this theoretical interpretation as those that do.

7) "Further research based on time-series analyses is required.." This seems too restrictive. Several analytical frameworks not commonly referred to as "time-series analysis" (e.g. various flavours of multistate models) would also be appropriate. As written this could be mis-interpreted as saying that ARIMA models are the only way to go here.

8) Subsection “Sampling bias control” "prevalence" or incidence?

9) Subsection “Sampling bias control” third paragraph "from a same sample"

10) Fourth paragraph , shouldn't "M_6_" be "M_7_"?

11) Final paragraph in subsection “Sampling bias control”. The logic of this is not clear. This seems to be saying that the incidence control value (A) is correlated with sampling effort (B), and that A is also correlated with antibiotic usage (C), and this will lead to spurious correlation between incidence and antibiotic use in unadjusted models. This may be correct (if we interpret spurious correlation to mean correlation between two variables in the absence of a direct causal relationship), but it is not clear from the text why it should be so. Can this be rephrased to make it clearer. A DAG may also help here (see, for example, Judea Pearl's The Book of Why for an introduction to DAGs in representing causal relationships) as unless there is a backdoor path, there won't be "spurious correlation".

12) Is an R^2^ of 0.34 really "strong correlation"?

13) Subsection “Connectivity and other ward characteristics”. The coding of fragility is confusing. The text says that an ordinal scale was used implying that it is treated as a categorical variable but with a natural ordering (but with no values associated with the distances between levels). However, it is then reported that numerical values were assigned to the levels. If it is an ordinal scale, the coding is arbitrary, but, as written, it sounds as though a numeric scale might actually have been used here. This needs to be clarified, and if a numeric scale has been used (i.e. the assumption has been made that patients in intensive care units have precisely twice the "fragility" as those in intermediate care units ) this either needs to be justified or (in the absence of clear rationale) the analysis revised, e.g. treating ward type as a categorical variable and not making any strong assumption in how patients differ in their vulnerability to infection.

14) "…exponentiated coefficients can be interpreted as the percentage increase of the incidence for every doubling of the predictor, when all other predictors are held at their reference value". Taken literally, this sentence is not accurate, even in the special case where the independent variable is log2 of the predictor. The sentence after this one is accurate and has all the information needed, so deleting the above sentence would be fine. Note that it is more usual to report results of a Poisson regression as incidence rate ratios (exponentiating the regression coefficients) which have a simple interpretation and will be familiar to many readers, and it's not clear why the conversion reported in the second sentence is preferred here.

15) Subsection “Pooled analysis of CTX/CRO- and IPM/MEM-resistant variants” The sentence beginning "The rationale.…" is very hard to make sense of, and we strongly suggest it is rewritten to make the meaning clearer.

16) The authors removed the ESKAPE2 acronym from the Abstract. It could also be removed from the conclusion in a similar way.

---

## [Author Response]

Essential revisions:Currently the description of the methods is incomplete and makes it difficult to assess the validity of the results. Specifics are described below.1) Is this incidence control value (Incidence(Variant|Ward) ) calculated for each of the 357 wards, or for each of the 3 ward types? This should be made clearer in the text, and also by introducing indexing into the mathematical expressions.

We agree that the section describing the incidence control value lacked clarity. We completely rewrote this section, using indexing of expressions where applicable. The calculation method was also slightly modified to take into account comment Q2 of the reviewers.

2) The authors build a method to account fori) variants being associated with diff anatomical sites P(Variant|Location)ii) different wards differing in which anatomical sites are typically sampled, and in how many samples are typically taken per patient P( No Variant | N, Ward )From the definition of P( No Variant | N, Ward ) it is not clear whether the authors also account for the fact that samples from one patient are not independent (in terms of the sampled anatomical site) and a definition of N(I,Ward) is missing. Is this the number of samples specific from a patient with index i, being disaggregated into location and ward, or is it the total number of samples in the ward disaggregated into the different anatomical sites? The difference is important: if N(I,ward) is the total number of ward samples disaggregated into anatomical sites, then the expression P(No Variant|N, Ward) assumes that samples taken from a patient i are independent.

We thank the reviewers for pointing out the lack of clarity regarding our simplifying assumptions. In the original version of our analysis, we assumed independence of the samples from a same patient in terms of both the probability of detection of a pathogen variant and the probability of being sampled from one anatomical site if already sampled from another site. Although the first independence assumption remained necessary to obtain tractable expressions, the second assumption was unnecessary and could be relaxed by considering the combinations of samples taken from a same patient.

We improved our analysis of the incidence control value by considering the probability that a patient is detected positive for a given variant, considering all samples taken in combination, rather than considering anatomical sites separately. Consequently, the Materials and methods section describing the incidence control value has been completely rewritten and all results of models involving the incidence control value have been updated. The simplifying assumptions are explicit in text. Noteworthy, this change of the computation of the incidence control value had a negligible impact on the results of our models and did not impact any of our findings. This suggests that the additional simplifying assumption in the original version of the manuscript was acceptable; we believe, however, that relaxing this assumption in the revised version strengthens the rigor of our approach.

3) In Data collection and compilation the authors say that clinical samples are deduplicated, resulting in one sample per patient, but in the section after it seems that multiple samples per patient are used for to compute incidence control. It is correct that for the incidence control value samples where used without deduplication, but for the regression model fit were deduplicated counts used? This should be made clearer.

Again, we thank the reviewers for this opportunity to clarify our methods. The deduplication of the clinical samples considered each unique combination of pathogen variant, patient and ward, so that a same patient can be counted multiple times if sampled in several wards or positive with several variants – but a same patient in a same ward with a same pathogen is counted only once. This deduplication was not applied when computing the incidence control value because we feel it is necessary to account for repeated sampling of the same patient, which increases the probability of pathogen detection. We clarified the text to describe the deduplication scheme used for observed incidence, and to explain why deduplication was not used to compute the incidence control value.

4) Using a priori expected selective antibiotic-variant pairs the authors show that the model manages to pick up more of those pairs than others. This is expected: even when substantial confounding increases all individual antibiotic-variant associations, a direct causal effect in a abx_A_-variant_X_ pair is expected to make it easier to pick up this association. It is not clear from the Materials and methods whether a multivariate model was used here. Please provide the regression equation in the Materials and methods to allow the reader to follow what was done. If it is univariate, it is to be expected that looking into different individual antibiotics will introduce a lot of new confounding from association with other antibiotic treatments. What would be useful here is a multivariate analysis of individual antibiotics (asking about the effect of each, given the effect of all the other treatment), and importantly including a few additional predictors (treatments) that one would expect to NOT be correlated with AMR (negative control). A zero-centred distribution of the negative control β parameters would then indicate lack of confounding.

We apologize for not making clear that all models were multivariate. We rewrote the Materials and methods section to clarify the model used (a quasi-Poisson model, see our response to comment Q5) and to provide a regression equation with the meaning of the overdispersion parameter. We also modified the legend of Figure 2 to clarify the multivariate nature of the analysis.

We fully agree with the reviewers on the interpretation of a zero-centered distribution of negative control coefficients as indicating negligible residual confounding. This interpretation was implicit in the construction of Figure 2, which clearly shows that the coefficients of associations not considered as “selective”, interpreted as negative controls, have a zero-centered distribution, contrary to the coefficients of possibly selective associations. We added a sentence in the Results section line 236-237 to clarify this interpretation.

5) A Poisson model was used. Typically Negative Binomial models will fit these type of data better as there's usually some overdispersion. The effect will be wider confidence intervals. It would be helpful to compare fits of Poisson and Neg Bin and go with the best fitting model.

Overdispersion was absent from most models and at most moderate (<3-fold) in some of them, which is why we reported Poisson regression results in the original version of the manuscript. For the sake of rigor, however, we examined under- and overdispersion in the models of Figure 1 by comparing the log-likelihoods of Poisson vs. negative binomial models and by examining the overdispersion parameter optimized under a quasi-Poisson model. This analysis is available in script F04_S1_overdispersion.R in the github repository. We include these results in Supplementary file 1 (Table 1B) and mention them the Materials and methods but we refrained from including them in details in the main manuscript. This analysis indicates: (1) that negative binomial models improved likelihood in few cases; and (2) that underdispersion (theta < 1) was present in several models and that its maximal magnitude was stronger than that of overdispersion (minimal theta 0.24 vs maximal theta 2.5).

To avoid any risk of underestimating the width of confidence intervals, we took dispersion into account in all models of the revised manuscript. Based on the findings above, we used quasi-Poisson rather than negative binomial models because: (1) underdispersion, which was prevalent, is better fit by quasi-Poisson models than by negative binomial models which are overdispersed by design; and (2) more importantly, negative binomial models weight towards the smaller sites (wards) whereas a quasi-Poisson weights toward the larger sites. As we consider the effects of wards with higher incidence to be more relevant to our scientific question than those of wards with lower incidence, it is more appropriate in this case to weight towards ward with greater incidence and to use quasi-Poisson models.

We modified the Materials and methods section accordingly and all results of regression models, including all figures, and Table 3. The changes did not modify any of our conclusions.

6) The repeated dichotomising between "significant" and "non-significant findings" in the paper is at odds with current statistical thinking (see, for example, the American Statistical Association consensus statement on p-values https://www.amstat.org/asa/files/pdfs/P-ValueStatement.pdf). We strongly suggest the authors drop this terminology and instead of reporting results as significant or not, report point estimates and confidence intervals for everything.

We agree with the reviewers on this point and, on a more personal note, we appreciate being encouraged (for the second occasion this year) to abandon P-values whose increasingly formal and rigid interpretation, especially in the medical literature, has diverged from their original intended use as an informal index. We have replaced all references to “significance” and P-values with confidence intervals in text. For readability, we conserved a Mann-Whitney test significance result in Figure 2C.

7) The exclusion of rarely used antibiotics from the analysis makes sense, but it wasn't clear why narrow spectrum antibiotics or those used in combination were also excluded. We think including these in the analysis would improve the paper.

Thank you for this suggestion. Some antibiotics had been excluded to lower the risk of multicollinearity in multivariate models in preliminary analyses with a smaller dataset. However, including them in analyses of the final, larger dataset did not hurt the models, so we are happy to include them. We have revised our analyses to include narrow spectrum antibiotic and those used in combination (specifically amoxicillin and aminoglycosides). We have updated the Materials and methods and Results accordingly, as well as Figure 2 and Table 3. This addition does not modify our conclusions.

8) There was a consensus that the conclusions were too strong e.g. "Our data establish that.…". As these are observational data, with lots of unmeasured confounding likely, the conclusions should better reflect the uncertainty.

We agree with this point. We have edited our manuscript throughout (particularly the Abstract and Discussion) to better reflect the uncertainty of our conclusions.

9) "Hospital-based antibiotic stewardship is a cornerstone strategy to combat AMR, based on the assumption that AMR evolves in hospitals. Yet, there is surprisingly limited evidence to support this assumption^47-49^"It's not completely clear what this statement means (there are lots of examples where antibiotic use in hospitalised patients clearly selects for resistant variants e.g. fusidic acid resistance in staphylococci), but if it's referring to population wide selection mediated by antibiotic use there are several additional papers that could be cited ( eg several papers by José-María López Lozano and Timothy Lawes, such as their Nature Micriobiology 2019 paper or more recently Tom Crellen's 2019 paper on Klebsiella in eLife).

We thank the reviewer for pointing out this sentence, which was both unclear and not necessary. We have removed this sentence.

[Editors' note: further revisions were suggested prior to acceptance, as described below.]

Essential revisions:1) In many places the paper refers to multivariate (or bivariate) models. Though this terminology is widely misused, multivariate (still) means models with multiple dependent variables. Models with multiple independent variables but a single dependent variable are *multivariable* models. The models used here seem to be multivariable and not multivariate.

The reviewers are correct. We have changed “multivariate” to “multivariable” throughout the manuscript and the Supplementary File.

2) In most places the authors have rightly been cautious about reaching causal conclusions based on the associations found in the observational data. However there are still a small number of places where results are presented using causal language (i.e. language that many readers are likely to interpret as implying a direct causal relationship) that is not fully justified e.g "with every doubling of antibiotic use increasing incidence by 49%", "connectivity had a larger effect on the incidence". The manuscript would be improved by rephrasing results to report these as associations, and the discussion can then consider the plausibility of a causal interpretation (as has already been done in most cases).

We agree with the reviewers. We have rephrased our results to eliminate this causal language:

“The largest effect size was found in carbapenem-resistant *K. pneumoniae*, in which every doubling of antibiotic use predicted a 47% increase in incidence (95% confidence interval, 19 to 90%).”

“Connectivity better predicted the incidence of carbapenem-resistant infections (4.8%, 95% CI, -0.4 to 10.7%) compared with 3GC-resistant infections (1.0%, 95% CI, -2.3 to 4.5%), in line with the comparatively stronger association of connectivity with the incidence of individual carbapenem-resistant variants (Figure 1).”

“As a note of caution, our observation that piperacillin-tazobactam use predicted a higher incidence of 3GC- and carbapenem-resistant variants does not imply selection for acquired resistance through any specific mechanism such as carbapenemase production.”

“We also note that the link between piperacillin-tazobactam use and global resistance resulted from the accumulation of small, positive associations with most 3GC- and carbapenem-resistant variants, including those with intrinsic resistance (Figure 2).”

3) "This nearly zero-centered distribution of coefficients in associations not expected to represent direct selection suggests that residual confounding in the models was negligible.": We would suggest revising this sentence to make it easier to understand.

This sentence required more space for clarity. To avoid interrupting the flow of the Results section with a detailed explanation, we moved this sentence in the Materials and methods section and rephrased it as follows,

“Of note, the associations not fulfilling the criteria for possible selection can be interpreted as negative controls in our models, and their coefficients are expected to be distributed around zero (null distribution) in the absence of residual confounding. In line with this interpretation, the coefficients of most of the negative control associations followed a near-zero-centered distribution (Figure 2C), suggesting that residual confounding was negligible in the adjusted models.”

4) "If the resistant infections in a ward mostly result from the admission of already colonized patients, antibiotic restrictions may have a limited impact on AMR compared to infection control measures that prevent the further dissemination of the pathogens" This makes sense if "limited impact on AMR" refers to colonisation with AMR (though it's not clear that it does given that the paper is concerned with clinical infection rather than carriage data), but antibiotics can also act to select for resistance within hosts increasing (or reducing) the chance of clinical infection with resistant organisms (see, for example, Niehus et al., 2020 and references therein). We strongly encourage the authors to rephrase to acknowledge that antibiotics can also have a big impact on within host dynamics and that antibiotic restrictions (or other changes) can also have an important impact even if most infections occur in already colonised patients.

We agree with this point and we should have avoided this ambiguous phrasing. We have revised this section of the Discussion:

“Understanding the respective impacts of antibiotic use and connectivity on the incidence of drug-resistant infections is essential for optimizing interventions against AMR. Pathogens whose incidence is strongly predicted by antibiotic use might be most effectively targeted by antibiotic restrictions. On the other hand, infection control interventions might be most relevant against pathogens whose incidence is predicted by connectivity. By applying a metapopulation framework to explain variations of infection incidences across a large network of hospital wards, we found that both antibiotic use and connectivity independently contribute to ward-level AMR in several pathogen species. Our study also provides the first quantitative ranking of the predicted impact of several key antibiotics on the global burden of drug-resistant infections in a hospital network.”

We thank the reviewers for pointing us to the Niehus et al. study which is highly relevant to our study setting. We modified the Introduction section to include this reference and better contextualize our approach.

“This is further complicated by the fact that the association between antibiotic use and resistance is not uniform across pathogen species (Bell et al., 2014) or classes of antibiotics (Niehus et al., 2020). Moreover, antibiotic consumption can also have long-term effects on the carriage of resistant bacteria within patients (Niehus et al., 2020).”

5) "the influence of connectivity on the incidence of a pathogen variant is expected to be higher if the variant is endemic to the hospital". This seems intuitive but if the level of endemicity is so high that all patients are colonised (as we would expect for E coli) then surely we would expect connectivity to have no effect on the incidence of infection with the pathogen variant. So is this statement generally true?

The reviewers are correct: if endemicity (interpreted as the concentration of the pathogen in the hospital relative to the community) is so high that all patients become colonized when entering the hospital, connectivity will not predict incidence anymore. Reasoning further, connectivity will not predict incidence whenever the proportion of colonized patients is constant across wards. Hence, some variability is required for connectivity to influence incidence. We have edited our statement to reflect this need for variation or heterogeneity across hospital wards:

“In our closed network model of the hospital, the influence of connectivity on the incidence of a pathogen variant is expected to be higher if the variant is endemic to the hospital, its prevalence varies across wards, and changes from one variant category to another are rare.”

6) "Consistent with this theoretical interpretation of connectivity, we found that its influence was strongest in the typical nosocomial pathogens P. aeruginosa and E. faecium." Though isn't it also the case that there was little evidence of an association between connectivity and MRSA infection incidence, but MRSA (or, at least, many lineages of MRSA) also represents a "typical nosocomial pathogen". It seems as important to highlight results that don't fit with this theoretical interpretation as those that do.

We agree with the reviewers. We wrongly used the term “nosocomial” in place of the more accurate term “hospital-endemic”, which better reflects our ecological reasoning. We do not believe that the case of MRSA is against our theoretical interpretation because although MRSA is considered a nosocomial pathogen, it is less clearly a hospital-endemic pathogen in our setting. We clarified this point and added references:

“Consistent with this theoretical interpretation of connectivity, we found that its influence was strongest in the hospital-endemic pathogens *P. aeruginosa* and *E. faecium* (Blanc et al., 2007; Wurster et al., 2016; Zhou et al., 2020). Intriguingly, MRSA incidence was not predicted by connectivity. This is at odds with the classical perception of MRSA as typically nosocomial and, by extension, hospital-endemic. However, available evidence does not strongly support the qualification of MRSA as a hospital-endemic pathogen in our setting. In France, the proportions of MRSA among *S. aureus* are comparable in community and hospital settings, at about 10% (ECDC, 2019; ONERBA France, 2018; Santé Publique France, 2019). MRSA population structures in hospitals and the community are comparably dominated by the so-called ST8 Lyon clone which is equally found in in- and outpatients (Dauwalder et al., 2008). Finally, MRSA infections were especially diffuse in our network, with a concentration index even lower than that of the less-resistant *S. aureus* infections (Table 1). Collectively, this does not support the conclusion that MRSA concentrates in French hospitals compared to the community, which might explain why connectivity did not predict MRSA incidence in our study.”

7) "Further research based on time-series analyses is required.." This seems too restrictive. Several analytical frameworks not commonly referred to as "time-series analysis" (e.g. various flavours of multistate models) would also be appropriate. As written this could be mis-interpreted as saying that ARIMA models are the only way to go here.

We agree with the reviewers. We have revised this sentence:

“Further research using, for instance, multi-state models or time-series analyses, may better clarify this ambiguity.”

8) Subsection “Sampling bias control” "prevalence" or incidence?

We have changed this to “incidence.”

9) Subsection “Sampling bias control” third paragraph "from a same sample"

We have edited this:

“For simplicity, we assumed independence between the pathogen variants isolated from a same sample and between samples collected from a same patient.”

10) Fourth paragraph , shouldn't "M_6_" be "M_7_"?

Thank you for pointing out this mistake. We corrected the text

11) Final paragraph in subsection “Sampling bias control”. The logic of this is not clear. This seems to be saying that the incidence control value (A) is correlated with sampling effort (B), and that A is also correlated with antibiotic usage (C), and this will lead to spurious correlation between incidence and antibiotic use in unadjusted models. This may be correct (if we interpret spurious correlation to mean correlation between two variables in the absence of a direct causal relationship), but it is not clear from the text why it should be so. Can this be rephrased to make it clearer. A DAG may also help here (see, for example, Judea Pearl's The Book of Why for an introduction to DAGs in representing causal relationships) as unless there is a backdoor path, there won't be "spurious correlation".

We agree that our rationale for the use of the incidence control value as an adjustment covariate should be clarified because it is central to our approach. We used DAGs to reason about the latent and observed variables during the initial model preparation steps but we refrained from presenting them in the original manuscript because we felt that the causal graph framework would not appeal to a clinical readership. However, we fully agree that the link between “sampling bias” and antibiotic use was not explicit and, as a result, was difficult to understand. Thus, we included a DAG as supplementary data to clarify our approach.

Indeed, the biasing (backdoor) path between the exposure (antibiotic use) and the outcome (incidence) is not directly related to “sampling bias” but to variations of the incidence and distribution of infections across wards. In a more formal causal reasoning, the “sampling bias” is a proxy variable to the distribution of infections rather than a variable directly on the biasing path.

To clarify this point, we added a DAG representation in Figure 1—figure supplement 2 and we added or rephrased the text in several places. We clarified early in text that the incidence control is a proxy to the distribution of infections, using the terminology of epidemiology:

“We also considered that the distribution of infections across wards was a source of bias that required a specific adjustment procedure (see Materials and methods). The local prevalence of specific infections (e.g., respiratory tract infections) in a ward influences both the antibiotic use and the observed incidence of infections with a given pathogen, which might confound the relationship between antibiotic use and incidence (Figure 1—figure supplement 2). However, the distribution of infections would be difficult to represent as an adjustment covariate with a sufficiently small number of categories. We used a proxy method to circumvent this issue. We assumed that the distribution of infections directly influences the frequency and specimen types (e.g., respiratory vs. urinary tract specimens) of microbiological samples in each ward. Under this assumption, we replaced the unrepresentable distribution of infections with a proxy variable summarizing the distribution of microbiological samples. This proxy variable, which we refer to as the incidence control, was defined as the ward-level incidence of a pathogen variant predicted by patterns of microbiological sampling alone.”

In line with this more accurate explanation of our approach, we replaced all references to “sampling bias correction” with “inclusion of the incidence control covariate” to clarify that microbiological sampling was not the primary source of bias. The relevant Materials and methods section was renamed “Controlling for microbiological sampling patterns.” We detailed our causal assumptions and we amended the last paragraph of this section:

“Under our assumption that the incidence control is a valid proxy to the unrepresentable distribution of infections in each ward, the incidence control should correlate both with antibiotic use and the incidence of infections. Bivariate analyses confirmed that the incidence control correlated with the observed cumulative incidence of all bacteria (R² = 0.96, 95% CI, 0.95 to 0.96, Figure 1—figure supplement 3) and, to a lesser extent, with the total antibiotic use (R² = 0.34, 95% CI, 0.25 to 0.40, Figure 1—figure supplement 4).”

12) Is an R^2^ of 0.34 really "strong correlation"?

Thank you for pointing this. We have edited this sentence as follows:

“Bivariate analyses confirmed that the incidence control correlated with the observed cumulative incidence of all bacteria (R² = 0.96, 95% CI, 0.95 to 0.96, Figure 1—figure supplement 3) and, to a lesser extent, with the total antibiotic use (R² = 0.34, 95% CI, 0.25 to 0.40, Figure 1—figure supplement 4).”

13) Subsection “Connectivity and other ward characteristics”. The coding of fragility is confusing. The text says that an ordinal scale was used implying that it is treated as a categorical variable but with a natural ordering (but with no values associated with the distances between levels). However, it is then reported that numerical values were assigned to the levels. If it is an ordinal scale, the coding is arbitrary, but, as written, it sounds as though a numeric scale might actually have been used here. This needs to be clarified, and if a numeric scale has been used (i.e. the assumption has been made that patients in intensive care units have precisely twice the "fragility" as those in intermediate care units ) this either needs to be justified or (in the absence of clear rationale) the analysis revised, e.g. treating ward type as a categorical variable and not making any strong assumption in how patients differ in their vulnerability to infection.

The reviewers are correct. Although our intention was to treat patient fragility as an ordinal variable, a numeric scale was used in previous analyses. We do not assume that patients in intensive care have precisely twice the fragility as those in intermediate care. As the coefficients of ordinal variables can be difficult to interpret, we have revised our analysis treating ward type as a categorical variable as suggested. All models that include ward type as an explanatory variable have been rerun and the corresponding Figures 1 and 3A have been modified accordingly. We have also revised the Results and Materials and methods as necessary. Due to small sample size in some ward type strata, however, the coefficients for ward type could not be determined for some pathogen variants. We explain this point early in text:

“In several pathogen variants, namely carbapenem-resistant *E. coli*, carbapenem-resistant *A*. *baumannii*, and vancomycin-resistant *E. faecium*, the small sample size in one or several ward categories prevented the inclusion of ward type as a model covariate.”

Importantly, these changes do not affect our conclusions.

14) "…exponentiated coefficients can be interpreted as the percentage increase of the incidence for every doubling of the predictor, when all other predictors are held at their reference value". Taken literally, this sentence is not accurate, even in the special case where the independent variable is log2 of the predictor. The sentence after this one is accurate and has all the information needed, so deleting the above sentence would be fine. Note that it is more usual to report results of a Poisson regression as incidence rate ratios (exponentiating the regression coefficients) which have a simple interpretation and will be familiar to many readers, and it's not clear why the conversion reported in the second sentence is preferred here.

The reviewers are correct. We have deleted this sentence as suggested. Of note, we chose to convert the coefficients to percent changes in infection incidence because we felt this would be easier to interpret for readers, especially those who work in clinical settings.

15) Subsection “Pooled analysis of CTX/CRO- and IPM/MEM-resistant variants” The sentence beginning "The rationale.…" is very hard to make sense of, and we strongly suggest it is rewritten to make the meaning clearer.

We have edited this sentence for clarity:

“When resistance to CTX/CRO or IPM/MEM was not determined by design (such as 3GC resistance in 3GCREC) or by intrinsic resistance (such as 3GC resistance in *E. faecium*), variants were classified as resistant when the proportion of resistance in our setting was above 80%. This less-stringent resistance criterion, compared to the criterion used to determine possibly selective associations, was chosen to avoid the exclusion of variants that are mostly resistant to an antibiotic group, which would bias pooled analyses.”

16) The authors removed the ESKAPE2 acronym from the Abstract. It could also be removed from the conclusion in a similar way.

We have made this change.

“To conclude, the modeling of the incidence of infections with 7 major bacterial species and their drug-resistant variants in hospital wards using a metapopulation framework indicates that both antibiotic use and inter-ward connectivity may predict the burden of AMR in a variant-specific fashion.”